# SPARSE AUTOENCODERS DO NOT FIND CANONICAL UNITS OF ANALYSIS

**Patrick Leask**[*]
Department of Computer Science
Durham University
patrick.leask@durham.ac.uk

**Bart Bussmann**[*]
Independent
bartbussmann@gmail.com

**Michael Pearce**
Independent

**Joseph Bloom**
Decode Research

**Curt Tigges**
Decode Research

**Noura Al Moubayed**
Department of Computer Science
Durham University

**Lee Sharkey**
Apollo Research

**Neel Nanda**

## ABSTRACT

A common goal of mechanistic interpretability is to decompose the activations of neural networks into features: interpretable properties of the input computed by the model. Sparse autoencoders (SAEs) are a popular method for finding these features in LLMs, and it has been postulated that they can be used to find a *canonical* set of units: a unique and complete list of atomic features. We cast doubt on this belief using two novel techniques: SAE stitching to show they are incomplete, and meta-SAEs to show they are not atomic. SAE stitching involves inserting or swapping latents from a larger SAE into a smaller one. Latents from the larger SAE can be divided into two categories: *novel latents*, which improve performance when added to the smaller SAE, indicating they capture novel information, and *reconstruction latents*, which can replace corresponding latents in the smaller SAE that have similar behavior. The existence of novel features indicates incompleteness of smaller SAEs. Using meta-SAEs - SAEs trained on the decoder matrix of another SAE - we find that latents in SAEs often decompose into combinations of latents from a smaller SAE, showing that larger SAE latents are not atomic. The resulting decompositions are often interpretable; e.g. a latent representing "Einstein" decomposes into "scientist", "Germany", and "famous person". To train meta-SAEs we introduce BatchTopK SAEs, an improved variant of the popular TopK SAE method, that only enforces a fixed *average* sparsity. Even if SAEs do not find canonical units of analysis, they may still be useful tools. We suggest that future research should either pursue different approaches for identifying such units, or pragmatically choose the SAE size suited to their task. We provide an interactive dashboard to explore meta-SAEs: https://metasaes.streamlit.app/

## 1 INTRODUCTION

Mechanistic interpretability aims to reverse-engineer neural networks into human-interpretable algorithms (Olah et al., 2020; Meng et al., 2022; Geva et al., 2023; Nanda et al., 2023; Elhage et al., 2021). A key challenge of mechanistic interpretability is identifying the correct units of analysis — fundamental components that can be individually understood and collectively explain the network's function. Ideally, these units would be *unique*, with no variations (Bricken et al., 2023); *complete*, encompassing all necessary features (Elhage et al., 2022); and *atomic* or *irreducible*, indivisible into smaller components (Engels et al., 2024). We refer to a set of units with all of these properties as **canonical**.

---

[*]These authors contributed equally to this work.

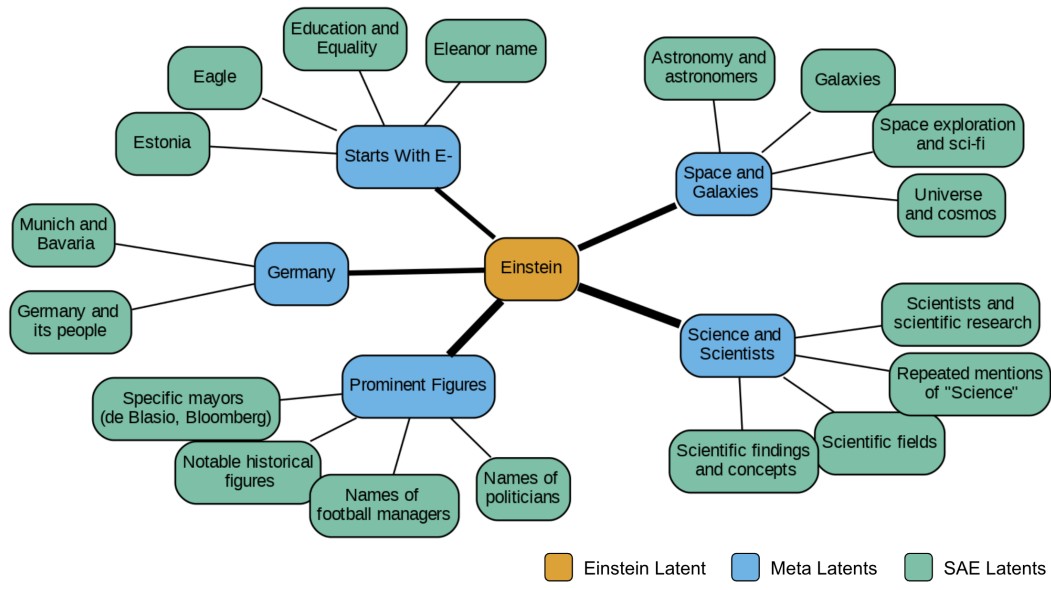

Figure 1: Decomposition of an SAE latent representing "Einstein" into a set of interpretable meta-latents. The edges connecting the nodes indicate shared activation by a meta-latent, with thicker lines representing stronger connections. It demonstrates the ability of meta-SAEs to uncover the underlying compositional structure of SAE latents, revealing how a complex concept can be represented as a sparse combination of meta-latents. We built a dashboard where you can explore all meta-latents: `https://metasaes.streamlit.app`

Initially, researchers hoped that individual MLP neurons (Meng et al., 2022; Olah et al., 2020) and attention heads (Wang et al., 2022; Olsson et al., 2022) could serve as these units. However, these proved insufficient for interpretability due to polysemanticity, where a single neuron responds to multiple unrelated concepts (Olah et al., 2020; Elhage et al., 2022).

Recently, sparse autoencoders (SAEs) have emerged as a promising alternative by decomposing the activations of LLMs into a dictionary of interpretable and monosemantic features (Bricken et al., 2023; Cunningham et al., 2023). A key hyperparameter when training SAEs is the dictionary size, i.e. number of latent units. Previous work conjectured that SAEs might identify a set of "true features" with sufficient dictionary size (Bricken et al., 2023), i.e. the canonical features that are the goal of much mechanistic interpretability research.

One challenge to the theory that SAEs identify a canonical set of units is the phenomenon of *feature splitting*, where latents from smaller SAEs "split" into multiple, more fine-grained latents in larger SAEs (Bricken et al., 2023). For example, Bricken et al. (2023) find a base64 feature that splits into three features in a larger SAE: activating on letters, digits, and encoded ASCII in base64 text. Furthermore, Templeton (2024) finds that a larger SAE has latents that activate on certain specific individual chemical elements that a smaller SAE did not represent. Currently, the effect of dictionary size on the features has not been systematically studied, in part because we lack good methods to compare latents found in SAEs of different sizes.

To better understand how SAEs of different sizes capture features, we develop a method called SAE stitching. When stitching SAEs, we systematically swap clusters of latents between SAEs of different dictionary sizes based on their cosine similarity. Through this method, we observe that larger SAEs learn both more fine-grained versions of latents found in smaller SAEs, but also entirely novel latents. The existence of novel latents suggests that the reconstruction error of smaller SAEs is partially due to missing out information altogether, not just imperfect approximations to features or overly coarse latents, indicating *incompleteness*.

Contrary to the story of feature splitting, we observed that some reconstruction latents had split from *multiple* latents in the smaller SAE, indicating those latents were composing into more complex latents. Previous work has predicted that the sparsity penalty incentivizes latents to represent

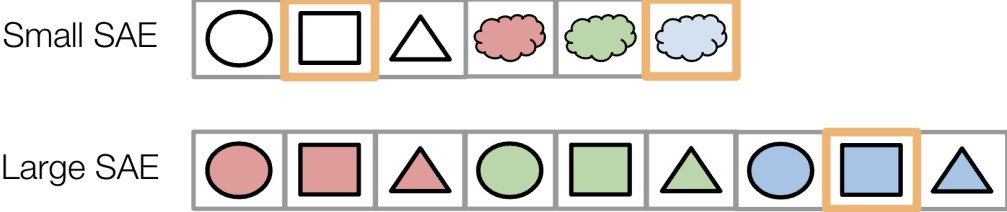

Figure 2: Example of composition of latents in SAEs of different sizes. The smaller SAE has six latents, three of which reconstruct shape features, and three of which reconstruct color features. Reconstructing a shape of a specific color requires two active latents (e.g. blue and square). On the other hand, the larger SAE has nine latents, each of which reconstructs a different color and shape combination. In the larger SAE, only a single active latent is required to reconstruct the colored shape (e.g. blue square). The sparsity penalty incentivizes larger SAEs to learn compositions of latents rather than atomic latents.

composed features, even if they are independent (Wattenberg & Viégas, 2024; Bricken et al., 2023; Anders et al., 2024). For instance, consider a neural network that represents color and shape features, each with three values (red/green/blue and circle/square/triangle). A small SAE might learn a latent for each value. A large SAE, however, might learn latents for all 9 color-shape combinations (i.e. blue square) instead of the 6 fundamental features (Smith, 2024). The large SAE can represent this is sparser as only one latent activates per input rather than two, see Figure 2.

To investigate this, we introduce meta-SAEs, which are SAEs trained to find sparse decompositions of the decoder directions of another SAE. These decompositions are often interpretable, e.g. a latent representing "Einstein" decomposes into meta-latents representing "scientist", "Germany", and "prominent figures", among others (see Figure 1). This shows that latents are often not *atomic*, especially in larger SAEs. We find that meta-latents are similar to latents in smaller SAEs, demonstrating that latents from larger SAEs can be interpreted as the composition of latents from smaller SAEs.

With SAE stitching, we observed reconstruction latents that had split from a small number of small SAE latents. To train Meta-SAEs, we wanted very few meta-latents active per input, e.g. four, a regime where the state-of-the-art JumpReLU SAEs (Rajamanoharan et al., 2024b) perform poorly. Another popular method is TopK SAEs (Gao et al., 2024), but these impose a fixed number of active meta-latents, which is also undesirable. To address these limitations we introduce BatchTopK, a novel method for training SAEs that enforces a fixed sparsity across a *batch* producing only a fixed *average* sparsity per individual input. We find that BatchTopK is useful for training SAEs on model activations too and is a Pareto improvement over TopK SAEs. Furthermore, it has better performance than JumpReLU SAEs at the desired sparsity levels for Meta-SAEs.

In summary, our contributions are:

1. **SAE stitching**, as a method for comparing latents across different sizes of SAE. Latents in a larger SAE are either novel latents, missing in smaller SAEs, or reconstruction latents, similar to some latents in smaller SAEs.

2. **Meta-SAEs**, as an approach for decomposing the decoder directions of SAEs into interpretable, monosemantic meta-latents.

3. We also introduce **BatchTopK SAEs**, the state-of-the-art SAE architecture for sparse dictionary learning on language model activations at a fixed average sparsity.

Our empirical results suggest that simply training larger SAEs is unlikely to result in a canonical set of units for all mechanistic interpretability tasks, and that the choice of dictionary size is subjective. We suggest taking a pragmatic approach to applying SAEs to mechanistic interpretability tasks, trying SAEs of several widths to see which is best suited. We are uncertain whether canonical units of analysis exist, but our results suggest that alternative approaches should be explored.

## 2 SPARSE AUTOENCODERS

Sparse dictionary learning is the problem of finding a decomposition of a signal that is both sparse and overcomplete (Olshausen & Field, 1997). Lee et al. (2007) initially applied the sparsity constraint to deep belief networks, with SAEs later being applied to the reconstruction of neural network activations (Bricken et al., 2023; Cunningham et al., 2023). In the context of large language models, SAEs decompose model activations $\mathbf{x} \in \mathbb{R}^n$ into sparse linear combinations of learned directions, which are often interpretable and monosemantic.

An SAE consists of an encoder and a decoder:

$$\mathbf{f}(\mathbf{x}) := \sigma(\mathbf{W}^{\text{enc}}\mathbf{x} + \mathbf{b}^{\text{enc}}), \tag{1}$$

$$\hat{\mathbf{x}}(\mathbf{f}) := \mathbf{W}^{\text{dec}}\mathbf{f} + \mathbf{b}^{\text{dec}}. \tag{2}$$

where $\mathbf{f}(\mathbf{x}) \in \mathbb{R}^m$ is the sparse latent representation and $\hat{\mathbf{x}}(\mathbf{f}) \in \mathbb{R}^n$ is the reconstructed input. $\mathbf{W}^{\text{enc}}$ is the encoder matrix with dimension $n \times m$ and $\mathbf{b}^{\text{enc}}$ is a vector of dimension $m$; conversely $\mathbf{W}^{\text{dec}}$ is the decoder matrix with dimension $m \times n$ and $\mathbf{b}^{\text{dec}}$ is of dimension $n$. The activation function $\sigma$ enforces non-negativity and sparsity in $\mathbf{f}(\mathbf{x})$, and a latent $i$ is active on a sample $\mathbf{x}$ if $f_i(\mathbf{x}) > 0$.

SAEs are trained on the activations of a language model at a particular site, such as the residual stream, on a large text corpus, using a loss function of the form

$$\mathcal{L}(\mathbf{x}) := \underbrace{\|\mathbf{x} - \hat{\mathbf{x}}(\mathbf{f}(\mathbf{x}))\|_2^2}_{\mathcal{L}_{\text{reconstruct}}} + \underbrace{\lambda \mathcal{S}(\mathbf{f}(\mathbf{x}))}_{\mathcal{L}_{\text{sparsity}}} + \alpha \mathcal{L}_{\text{aux}} \tag{3}$$

where $\mathcal{S}$ is a function of the latent coefficients that penalizes non-sparse decompositions, and $\lambda$ is a sparsity coefficient, where higher values of $\lambda$ encourage sparsity at the cost of higher reconstruction error. Some architectures also require the use of an auxiliary loss $\mathcal{L}_{\text{aux}}$, for example to recycle inactive latents in TopK SAEs (Gao et al., 2024). We expand on the different SAE variants in Appendix A.2 and provide a glossary of terms in Appendix A.1.

## 3 RELATED WORK

**SAEs for Mechanistic Interpretability.** SAEs have been demonstrated to recover sparse, monosemantic, and interpretable features from language model activations (Bricken et al., 2023; Cunningham et al., 2023; Templeton, 2024; Gao et al., 2024; Rajamanoharan et al., 2024a;b), however their application to mechanistic interpretability is nascent. After training, researchers often interpret the meaning of SAE latents by examining the dataset examples on which they are active, either through manual inspection using features dashboards (Bricken et al., 2023) or automated interpretability techniques (Gao et al., 2024). SAEs have been used for circuit analysis (Marks et al., 2024) in the vein of (Olah et al., 2020; Olsson et al., 2022); to study the role of attention heads in GPT-2 (Kissane et al., 2024); and to replicate the identification of a circuit for indirect object identification in GPT-2 (Makelov et al., 2024). Transcoders, a variant of SAEs, have been used to simplify circuit analysis and applied to the greater-than circuit in GPT-2 (Dunefsky et al., 2024). While these applications highlight SAEs as valuable tools for understanding language models, it remains unclear whether they identify canonical units of analysis.

**Representational Structure.** Language models trained on the next-token prediction task learn representations that model the generative process of the data. For example, Li et al. (2022) found that transformers trained by next-move prediction to play the board game Othello explicitly represent the board state; and Gurnee & Tegmark (2023) used linear probes to predict geographical and temporal information from language model activations. SAEs learn these structures as well, for example, the activations of a cluster of GPT-2 SAE latents form a cycle when reconstruction activations for weekday name tokens (Engels et al., 2024). Bricken et al. (2023) find evidence of convergent global structure by applying a 2-dimensional UMAP transformation to decoder directions of SAEs of different sizes. This results in a rich structure of latent projections with regions of latents relating to similar concepts close to each other.

**Tuning Dictionary Size.** Previous work on tuning dictionary size has mixed findings regarding how SAEs scale with dictionary size. For instance, Templeton (2024) observed that larger SAEs learn latents absent in smaller ones, such as specific chemical elements. Conversely, Bricken et al. (2023) found similar latents across various SAE sizes, noting that latents in smaller SAEs sometimes split into multiple latents as the dictionary size increases (Appendix A.4 includes such examples taken from our SAEs). Currently, the effect of dictionary size on the learned latents has not been systematically studied.

**Model Stitching.** Model stitching is a method by which layers from one neural network are "stitched" or swapped into another neural network. Lenc & Vedaldi (2015); Bansal et al. (2021) propose that model stitching can be used to quantify representation similarity across different neural network architectures and training regimes by demonstrating that if two networks trained on the same task can be stitched together with minimal loss in performance, it suggests a high degree of alignment in their intermediate representations.

## 4 SAE STITCHING

Templeton (2024) finds that a larger SAE has latents that activate on certain specific individual chemical elements that a smaller SAE did not represent. Conversely, Bricken et al. (2023) observes that smaller SAEs learn latents activating on broad mathematical text, while larger SAEs learn latents activating on more specific mathematical categories. On the one hand, this suggests that training large SAEs is necessary to learn latents relating to all relevant concepts. On the other hand, this means that some of the capacity of larger SAEs is used to represent similar information as smaller SAEs, but with more latents.

To compare latents of SAEs with different dictionary sizes, we introduce SAE stitching. In SAE stitching, we add or replace latents in one SAE with latents from another and observe the effect on the reconstruction performance. We find that larger SAEs learn both finer-grained versions of latents from smaller SAEs (*reconstruction latents*) and entirely new latents that capture additional information (*novel latents*). SAE stitching allows us to identify these two categories of latents in larger SAEs.

### 4.1 STITCHING OPERATION

The output of an SAE can be expressed as a sum of the contributions from the individual latents:

$$\hat{\mathbf{x}} := \sum_{i=0}^{d} \mathbf{W}_i^{\text{dec}} f_i(\mathbf{x}) + \mathbf{b}^{\text{dec}} \tag{4}$$

where $d$ is the size of the SAE dictionary, $\mathbf{W}_i^{\text{dec}}$ is an individual decoder direction, $f_i(\mathbf{x})$ is the activation value for the latent $i$, and $\mathbf{b}^{\text{dec}}$ is the decoder bias term. To stitch latents from one SAE into another, we modify the reconstruction as follows:

$$\hat{\mathbf{x}} := \alpha \mathbf{b}_0^{\text{dec}} + (1 - \alpha)\mathbf{b}_1^{\text{dec}} + \sum_{l_0 \in L_0} \mathbf{W}_{0,l_0}^{\text{dec}} f_{0,l_0}(\mathbf{x}) + \sum_{l_1 \in L_1} \mathbf{W}_{1,l_1}^{\text{dec}} f_{1,l_1}(\mathbf{x}) \tag{5}$$

where $\alpha = \frac{|L_0|}{|L_0| + |L_1|}$, $\mathbf{W}_{0,l_0}^{\text{dec}}$ and $\mathbf{W}_{1,l_1}^{\text{dec}}$ represent individual decoder directions and $L_0$ and $L_1$ are the set of latents we include from the respective SAEs. Unlike with model stitching (Bansal et al., 2021), SAE decoder directions are privileged and require no transformations to stitch.

We experiment with stitching on eight SAEs trained on the residual stream of layer 8 of GPT 2 Small (Radford et al., 2019) with dictionary sizes ranging from 768 to 98,304 and two of the Gemma Scope SAEs (Lieberum et al., 2024) trained on Gemma 2 2B (Team et al., 2024) with dictionary size 16,384 and 32,768. For the full list of SAE properties and training details see Appendix A.6.

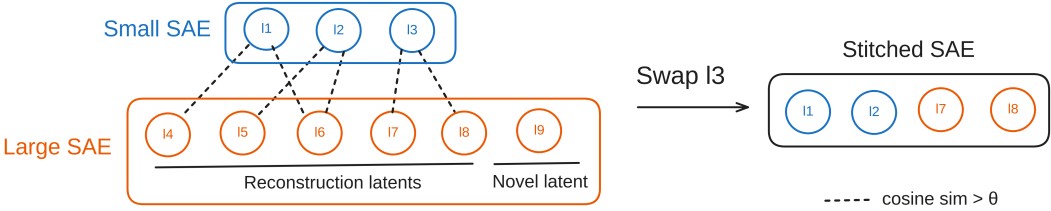

Figure 3: SAE stitching operation: connected subgraphs of latents can be swapped between SAEs based on cosine similarity

## 4.2 NOVEL AND RECONSTRUCTION LATENTS

Using SAE stitching, we want to find out whether latents in larger SAEs are just more fine-grained versions of latents of smaller SAEs, or whether they represent novel information that is missed by the smaller SAEs. If we stitch a latent from a larger SAE into a smaller SAE and the reconstruction performance improves, this indicates that the latent is adding new information to the smaller SAE. If the reconstruction performance worsens instead, this indicates that the latent is overlapping and the stitched SAE is now representing the same information twice. In this case, we can instead swap the latent from the smaller SAE with its corresponding latents from the larger SAE.

However, evaluating all combinations of latents when stitching two SAEs is computationally infeasible. Bricken et al. (2023) finds that the decoder directions of latents with similar behavior across SAEs of different sizes form local clusters with high cosine similarity. As such, we propose thresholding on the maximum decoder cosine similarity metric as a heuristic to classify latents into the two categories, i.e. for an SAE decoder direction $\mathbf{W}_{1,i}^{\text{dec}}$, the feature is assigned to the novel group if

$$\max_j \left( \frac{\mathbf{W}_{1,i}^{\text{dec}} \cdot \mathbf{W}_{0,j}^{\text{dec}}}{\|\mathbf{W}_{1,i}^{\text{dec}}\| \|\mathbf{W}_{0,j}^{\text{dec}}\|} \right) < \theta \tag{6}$$

where $\mathbf{W}_{0,j}^{\text{dec}}$ represents any decoder direction in $\mathbf{W}_0^{\text{dec}}$. If the maximum cosine similarity is larger than the threshold, it is assigned to the reconstruction group. Figure 4 shows the relationship between cosine similarity and the effect on the reconstruction loss of independently stitching that feature for a pair of GPT-2 SAEs. In our experiments we set the cosine threshold to 0.7 for the GPT-2 SAEs, and 0.4 for the Gemmascope SAEs, based on our exploratory analysis. Appendix Figure 15 shows the ROC curve for using different thresholds to predict the effect of adding a latent based on the maximum cosine similarity.

Inserting reconstruction latents into the target SAE decreases the performance due to overlapping functionality. We resolve this by removing latents in the target SAE that have a high cosine similarity to the ones we are inserting, effectively swapping them with similar latents. To select which latents we swap, we construct a bipartite graph where latents from the smaller SAE form one set of vertices and latents from the larger SAE form the other. We connect latents if they have a decoder cosine similarity above the threshold. For each connected subgraph, the latents from the smaller SAE can be swapped by their corresponding connected latents in the larger SAE (see Appendix A.5 for examples of connected subgraphs).

Figure 5 shows four transitions between pairs of SAEs of increasing size. For each transition, we first identify novel latents from the larger SAE and add them one by one, increasing the average L0 since we are inserting extra latents. We then identify groups of related reconstruction latents as defined by our bipartite graph and swap each group - removing similar latents from the smaller SAE and replacing them with their counterparts in the larger SAE. This decreases L0 on average since the larger SAE represents similar information more sparsely. Each step leads on average to improved reconstruction performance, allowing us to interpolate between different SAEs in terms of dictionary size, sparsity, and reconstruction performance.

The existence of the novel group of latents demonstrates that smaller SAEs are *incomplete*, and that larger SAE learn features that are missed by the smaller SAE. The reconstruction group seems to largely reconstruct the same features as the latents in the smaller SAE, but uses more features to

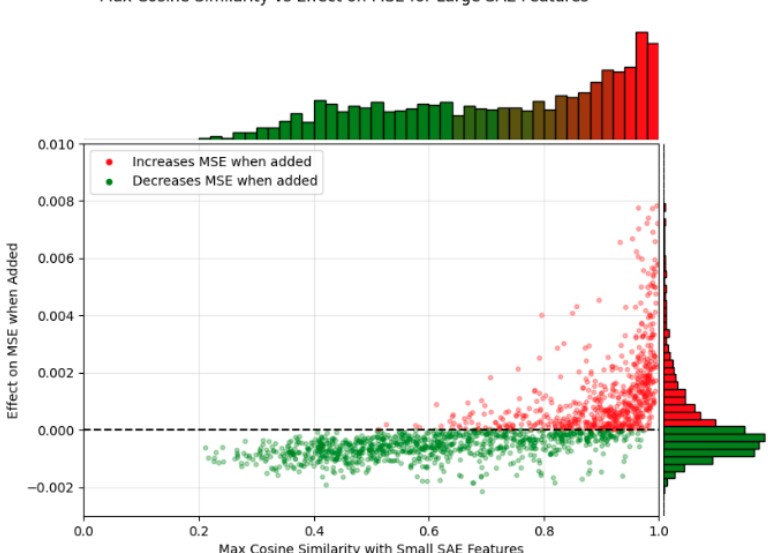

Figure 4: Change in MSE when adding each feature from GPT2-1536 to GPT2-768, plotted against the maximum cosine similarity of that feature to any feature in GPT2-768. Features with cosine similarity less than 0.7 tend to improve MSE, while more redundant features hurt performance. A few extreme outliers with very high cosine similarity and effect on MSE are not visible in this plot.

achieve lower sparsity (Appendix Figure 16). In some cases, a single latent splits into two more specific latents - this is the feature splitting observed in Bricken et al. (2023). However, we find some reconstruction group latents have high decoder cosine similarity to *multiple* latents in the smaller SAE, suggesting the large SAE latent is an interpolation or composition of the smaller SAE latents (Appendix A.5). This supports the predictions of Wattenberg & Viégas (2024); Bricken et al. (2023); Anders et al. (2024) that the sparsity penalty results in the undesirable composition of features that may be sparser, but do not add any new information. We explore this phenomenon further in Section 5.

# 5 META-SAES

In Section 4, we demonstrated through SAE stitching that increasing dictionary size leads to larger SAEs learning not only novel features, but also reconstruction latents that encode similar information to the latents in smaller SAEs. We found that some of these reconstruction latents have high cosine similarity with *multiple* latents in the smaller SAE, see Appendix A.5. This suggests that these smaller SAE latents are composing into more complex latents, such as in the example of a latent representing "blue" and another latent representing "square" combining in a "blue square"-latent (see Figure 2). If large SAE features are indeed compositions rather than *atomic*, it may be possible to decompose them into more fundamental units.

To decompose the latents of larger SAEs, we introduce meta-SAEs. Meta-SAEs are SAEs trained to reconstruct the decoder directions $\mathbf{W}_i^{\mathrm{dec}}$ of a standard SAE using a dictionary of meta-latents, rather than reconstructing network activations. That is, we treat the latents $\mathbf{W}_i^{\mathrm{dec}}$ as the training data for our meta-SAE. We find that meta-latents and meta-SAE decompositions are interpretable, with the meta-latents being monosemantic. Table 1 provides some examples of meta-SAE decompositions.

The latents of meta-SAEs have similar decoder directions to those found in SAEs of comparable size trained directly on the same network activations. This observation further supports the hypothesis that larger SAE latents are not entirely new features but may be compositions of features already learned, albeit less precisely, by smaller models.

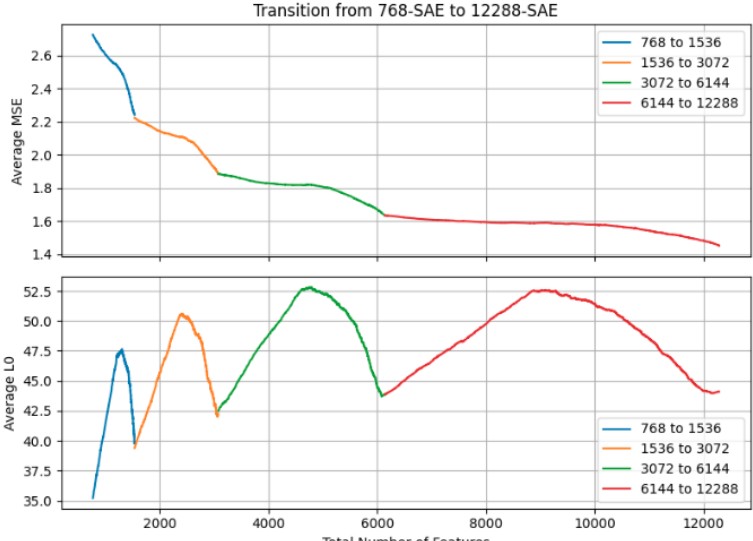

Figure 5: Interpolating between SAE pairs of increasing dictionary size (768→1536→3072→6144→12288) through two steps per phase: adding novel latents (increasing L0) then swapping groups of reconstruction latents (decreasing L0 on average). Both steps on average improve reconstruction (MSE). The L0 and MSE are averages over input samples.

Table 1: Example GPT-2 SAE latents and their meta-latent decompositions, with model-generated explanations (human-summarized) of what the latents activate on. More latent decompositions can be found on our interactive dashboard: `https://metasaes.streamlit.app`

| SAE Latent Description | Meta-Latent Descriptions |
|---|---|
| Albert Einstein | Science & Scientists, Famous People, Space & Astronomy, Germany, Electricity, Words starting with a capital E |
| Rugby | Sports activities, Words starting with 'R', References to Ireland, References to sports leagues, activities & actions |
| Android Operating System | Mobile phones, operating systems, Californian cities |

We use the BatchTopK SAE, as described in Section 6, to train our meta-SAEs. We train our meta-SAE on the decoder directions of the GPT-2 SAE with dictionary size 49152. The meta-SAE has a dictionary size of 2304 meta-latents, with on average 4 of meta-latents active per SAE latent. Due to small number of training samples for the meta-SAE (49152), the meta-SAE is trained for 2000 epochs. We use the Adam optimizer with learning rate 1e-4 and a batch size of 4096. After training, the meta-SAE explains 55.47% of the variance of the decoder directions of the SAE.

## 5.1 EVALUATING META-SAE DECOMPOSITIONS

We follow the lead of (Bills et al., 2023; Bricken et al., 2023; Cunningham et al., 2023; Rajamanoharan et al., 2024b) in evaluating neural network and SAE latents using automated interpretability with LLMs.

First, we generate explanations of SAE latents by presenting GPT-4o-mini with a list of input sequences that activate an SAE latent to varying degrees, and prompting it to generate a natural language explanation of the feature consistent with the activations. Second, we collect all of the SAE latents on which a meta-SAE latent is active, and prompt again with the explanation and a number of top activating examples of each of the SAE latents, asking the model to provide an explanation of the common behavior of the SAE latents, which becomes the meta-SAE latent explanation.

We evaluate the meta-SAE latent explanations in a zero-shot multiple-choice-question setting. For a given latent we prompt GPT-4o-mini with the explanations of the meta-SAE latents that are active on

that latent, and ask it to choose which of 5 SAE latent explanations most relate to the explanations of the meta-SAE latents. One of these SAE latent explanations is of the correct latent, with the remaining 4 explanations corresponding to random latents from the SAE. On a random sample of 1,000 SAE latents, GPT-4o-mini chose the correct answer of the five options 73% of the time.

## 5.2 COMPARISON TO SMALLER SAE LATENTS

In Section 4, we hypothesised that latents in larger SAEs can be described as the composition of latents in smaller SAEs. We find that meta-SAEs learn similar latents to similar sized SAEs trained on the original reconstruction problem. Plots of the maximum cosine similarity between meta-SAE latents and latents from SAEs of different sizes are shown in Figure 22. We validate this by replacing meta-SAE decoder directions with the most similar SAE decoder direction, and retrained the encoder. This results in only a small decrease in meta-SAE reconstruction performance (Appendix Figure 23). This suggests that larger SAE latents are indeed composed of latents from smaller SAEs.

## 6 BATCHTOPK SPARSE AUTOENCODERS

In Section 5, we hypothesised that SAE latents will linearly decompose into a small, variable number of meta-latents. Of the popular SAE variants, top-k SAEs (Gao et al., 2024) decompose each input into a fixed number of latents, whilst ReLU (Bricken et al., 2023; Cunningham et al., 2023) and JumpReLU Rajamanoharan et al. (2024b) have poor performance at that sparsity level and require hyperparameter tuning to achieve a specific sparsity level. We provide further details on these SAE variants in Appendix A.2.

To address these limitations, we introduce BatchTopK SAEs, a novel variant that extends the TopK concept to operate at the batch level rather than on individual samples. This approach allows for a flexible number of active latents per sample while maintaining a desired average sparsity across the batch. Therefore, we use BatchTopK SAEs throughout our meta-SAE experiments.

The training objective for BatchTopK SAEs is defined as:

$$\mathcal{L}(\mathbf{X}) = \|\mathbf{X} - \text{BatchTopK}(\mathbf{W}^{\text{enc}}\mathbf{X} + \mathbf{b}^{\text{enc}})\mathbf{W}^{\text{dec}} + \mathbf{b}^{\text{dec}})\|_2^2 + \alpha\mathcal{L}_{\text{aux}} \tag{7}$$

Here, $\mathbf{X}$ is the input data batch; $\mathbf{W}^{\text{enc}}$ and $\mathbf{b}^{\text{enc}}$ are the encoder weights and biases, respectively; $\mathbf{W}^{\text{dec}}$ and $\mathbf{b}^{\text{dec}}$ are the decoder weights and biases. The BatchTopK function sets all activation values to zero that are not among the top $n \times k$ activations by value in the batch, not changing the other values. The term $\mathcal{L}_{\text{aux}}$ is an auxiliary loss scaled by the coefficient $\alpha$, used to prevent dead latents, and is the same as in TopK SAEs.

BatchTopK introduces a dependency between the activations for the samples in a batch. We alleviate this during inference by using a threshold $\theta$ that is estimated as the average of the minimum positive activation values across a number of batches:

$$\theta = \mathbb{E}_{\mathbf{X}}[\min\{z_{i,j}(\mathbf{X})|z_{i,j}(\mathbf{X}) > 0\}] \tag{8}$$

where $z_{i,j}(\mathbf{X})$ is the $j$th latent activation of the $i$th sample in a batch $\mathbf{X}$. With this threshold, we use the JumpReLU activation function during inference instead of the BatchTopK activation function, zeroing out all activations under the threshold $\theta$.

We evaluate the performance of BatchTopK on the activations of two LLMs: GPT-2 Small (residual stream layer 8) and Gemma 2 2B (residual stream layer 12). We use a range of dictionary sizes and values for $k$, and compare our results to TopK and JumpReLU SAEs in terms of normalized mean squared error (NMSE) and cross-entropy degradation. For experimental details, see Appendix A.3.

We find that for a fixed number of active latents (L0=32) the BatchTopK SAE has a lower normalized MSE and less cross-entropy degradation than TopK SAEs on both GPT-2 activations (Figure 6) and Gemma 2 2B (Figure 7). Furthermore, we find that for a fixed dictionary size (12288) BatchTopK outperforms TopK for different values of k on both models.

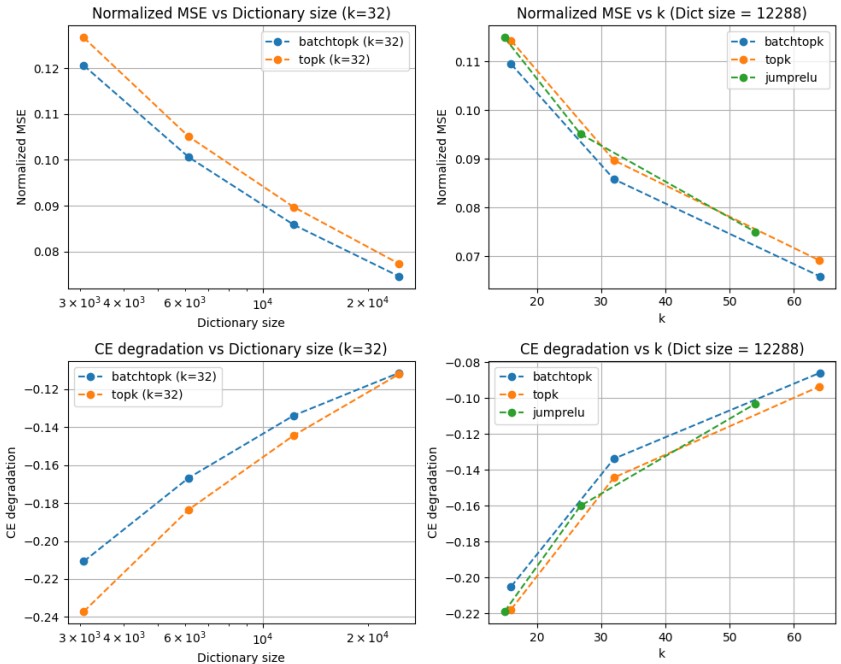

Figure 6: On GPT-2 Small activations, BatchTopK largely achieves a better NMSE and CE compared to standard TopK across different dictionary sizes, for a fixed number of active latents of 32 (Left). JumpReLU SAEs are omitted from this comparison as their L0 cannot be fixed to a value. For fixed dictionary size (12,288) and different levels of k, BatchTopK outperforms TopK and JumpReLU SAES, both in terms of NMSE and CE (Right).

In addition, BatchTopK outperforms JumpReLU SAEs on both measures on GPT-2 Small model activations at all levels of sparsity. On Gemma 2 2B model activations the results are more mixed: although BatchTopK achieves better reconstruction than JumpReLU for all values of k, BatchTopK only outperforms JumpReLU in terms of CE degradation in the highest sparsity setting (k=16).

# 7 CONCLUSION

Our findings challenge the idea that SAEs can discover a canonical set of features. Through SAE stitching, we demonstrated that smaller SAEs are incomplete, missing information that novel features in larger SAEs capture. Moreover, our meta-SAE experiments showed that, due to the sparsity penalty, latents in larger SAEs are often not atomic but compositions of interpretable meta-latents. These findings suggest that there is no single SAE width at which it learns a unique and complete dictionary of atomic features that can be used to explain the behavior of the model.

These results imply that rather than converging on a unique, complete, and irreducible set of features, SAEs of different sizes offer varying granularities and compositions of features. This indicates that the choice of SAE size should be guided by the specific interpretability task at hand, accepting that no single SAE configuration provides a universal solution. However, our methods neither identify canonical units of analysis, nor the size of dictionary to use for a given task. Furthermore, our work only studies two LLMs and does not include very large SAEs, such as in Templeton (2024). We also acknowledge that the use of SAEs in mechanistic interpretability is nascent, and whilst early results are encouraging, associating the learned latents of SAEs with interpretable concepts is still an open problem. In conclusion, our research suggests that alternative methods are required for identifying canonical units, and that SAE practitioners should embrace a pragmatic approach towards choosing dictionary size when using SAEs on interpretability tasks such as probing, unlearning and steering.

ACKNOWLEDGEMENTS

We thank the ML Alignment and Theory Scholars (MATS) program for facilitating this work. In particular, we are grateful for McKenna Fitzgerald and Matthew Wearden for their support in planning and managing this collaboration. We are also thankful for the helpful discussions we had with Adam Karvonen, Can Rager, Javier Ferrando, Oscar Obeso, Stepan Shabalin, and Slava Chalnev about this work. Finally, BB and PL want to thank AI Safety Support for funding this work.

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

## A    APPENDIX / SUPPLEMENTAL MATERIAL

### A.1    GLOSSARY OF TERMS

**Active Latents (L0):**    For an input $x$ and SAE activation function $f(x)$, the number of non-zero elements in $f(x)$. Typically measured as average L0 across a batch: $L0 = \frac{1}{n} \sum_i \|f(x_i)\|_0$.

**BatchTopK:**    A sparsification method that selects the top $n \times k$ activations across a batch of size $n$, rather than $k$ activations per sample, maintaining a fixed average sparsity. For a $n \times m$ batch $\mathbf{X}$, BatchTopK$(\mathbf{X})_{ij} = \mathbf{X}_{ij}$ if $\mathbf{X}_{ij}$ is among $n \times k$ largest elements, 0 otherwise.

**Canonical Unit:**    Hypothetical, fundamental building blocks of a LLMs computation that are unique, complete, and atomic.

**Cross-Entropy Degradation:**    The increase in cross-entropy loss when replacing the model activations with the reconstruction of the SAE.

**Decoder Directions:**    The columns of the decoder matrix $W^{\text{dec}}$ that map from latent to input space. Two decoder directions with high cosine similarity suggest related features.

**Dictionary Size:**    The dimensionality of the latent space in an SAE, determining the maximum number of unique features that can be learned.

**Feature Splitting:**    A phenomenon where a broad latent learned by a smaller SAE splits into more fine-grained latents in a larger SAE.

**Latent:**    The encoder-decoder pair corresponding to single element in the SAE's dictionary, i.e. a learned feature of the SAE rather than a feature of the data.

**Mechanistic Interpretability:**    The study of reverse-engineering neural networks into interpretable algorithms, focusing on identifying and understanding computational features and circuits.

**Meta-latents:**    Features learned by a meta-SAE when trained on the decoder directions of another SAE.

**Monosemantic:**    Property of a feature that responds selectively to a single coherent concept. Contrasts with polysemantic features.

**Novel Latents:**    Features in a larger SAE with maximum cosine similarity below threshold $\theta$ to any feature in a smaller SAE, indicating capture of previously unrepresented information.

**Polysemanticity:**    The phenomenon where individual neurons or features respond to multiple unrelated concepts.

**Reconstruction Latents:** Features in a larger SAE with maximum cosine similarity above threshold $\theta$ to features in a smaller SAE, representing refined or specialized versions of existing features.

**Residual Stream:** In the context of transformer architectures, the residual stream refers to the main information flow that bypasses the self-attention and feed-forward layers through residual connections.

**SAE Stitching:** A technique for analyzing feature relationships across SAEs of different sizes by systematically transferring latents based on decoder similarity.

**Sparsity Coefficient ($\lambda$):** Hyperparameter in the loss function of some SAE architectures $L = \|x - \hat{x}\|^2 + \lambda S(f(x))$ controlling the trade-off between reconstruction accuracy and activation sparsity.

**TopK:** A sparsification approach that maintains exactly $k$ non-zero activations per input by zeroing all but the $k$ largest values: $\text{TopK}(x)_i = x_i$ if $x_i$ is among $k$ largest elements, 0 otherwise.

## A.2 SAE VARIANTS

**ReLU SAEs** (Bricken et al., 2023) use the L1-norm $S(\boldsymbol{f}) := \|\boldsymbol{f}\|_1$ as an approximation to the L0-norm for the sparsity penalty. This provides a gradient for training unlike the L0-norm, but suppresses latent activations harming reconstruction performance (Rajamanoharan et al., 2024a). Furthermore, the L1 penalty can be arbitrarily reduced through reparameterization by scaling the decoder parameters, which is resolved in Bricken et al. (2023) by constraining the decoder directions to the unit norm. Resolving this tension between activation sparsity and value is the motivation behind more recent architecture variants.

**TopK SAEs** (Gao et al., 2024; Makhzani & Frey, 2014) enforce sparsity by retaining only the top $k$ activations per sample. The encoder is defined as:

$$\boldsymbol{f}(\boldsymbol{x}) := \text{TopK}(\mathbf{W}^{\text{enc}}\mathbf{x} + \mathbf{b}^{\text{enc}}) \tag{9}$$

where TopK zeroes out all but the $k$ largest activations in each sample. This approach eliminates the need for an explicit sparsity penalty but imposes a rigid constraint on the number of active latents per sample. An auxiliary loss $\mathcal{L}_{aux} = \|e - \hat{e}\|^2$ is used to avoid dead latents, where $\hat{e} = W^{dec}z$ is the reconstruction using only the top-$k_{aux}$ dead latents (usually 512), this loss is scaled by a small coefficient $\alpha$ (usually 1/32).

**JumpReLU SAEs** (Rajamanoharan et al., 2024b) replace the standard ReLU activation function with the JumpReLU activation, defined as

$$\text{JumpReLU}_\theta(z) := zH(z - \theta) \tag{10}$$

where $H$ is the Heaviside step function, and $\theta$ is a learned parameter for each SAE latent, below which the activation is set to zero. JumpReLU SAEs are trained using a loss function that combines L2 reconstruction error with an L0 sparsity penalty, using straight-through estimators to train despite the discontinuous activation function. A major drawback of the sparsity penalty used in JumpReLU SAEs compared to (Batch)TopK SAEs is that it is not possible to set an explicit sparsity and targeting a specific sparsity involves costly hyperparameter tuning. While evaluating JumpReLU SAEs, Rajamanoharan et al. (2024b) chose the SAEs from their sweep that were closest to the desired sparsity level, but this resulted in SAEs with significantly different sparsity levels being directly compared. JumpReLU SAEs use no auxiliary loss function.

### A.3   BATCHTOPK SAES

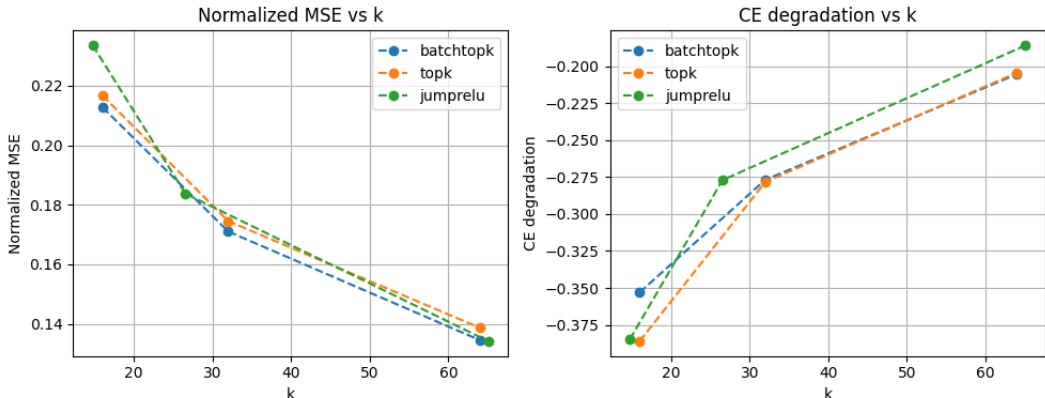

Figure 7: On Gemma 2 2B activations, BatchTopK outperforms TopK SAEs across different values of k. Although BatchTopK has a better reconstruction performance (left), it only outperforms JumpReLU in terms of downstream CE degradation in the setting where k=16 (right).

In this appendix, we provide details about the datasets used, model configurations, and hyperparameters for our experiments.

We trained our sparse autoencoders (SAEs) on the OpenWebText dataset[1], which was processed into sequences of a maximum of 128 tokens for input into the language models.

All models were trained using the Adam optimizer with a learning rate of $3 \times 10^{-4}$, $\beta_1 = 0.9$, and $\beta_2 = 0.99$. The batch size was 4096, and training continued until a total of $1 \times 10^9$ tokens were processed.

We experimented with dictionary sizes of 3072, 6144, 12288, and 24576 for the GPT-2 Small model, and used a dictionary size of 16384 for the experiment on Gemma 2 2B. In both experiments, we varied the number of active latents $k$ among 16, 32, and 64.

For the JumpReLU SAEs, we varied the sparsity coefficient such that the resulting sparsity would match the active latents $k$ of the BatchTopK and TopK models. The sparsity penalties in the experiments on GPT-2 Small were 0.004, 0.0018, and 0.0008. For the Gemma 2 2B model we used sparsity penalties of 0.02, 0.005, and 0.001. In both experiments, we set the bandwidth parameter to 0.001.

The results of our experiments on Gemma 2 2B can be found in figure 7.

To confirm that BatchTopK SAEs make use of the enabled flexibility to activate a variable number of latents per sample, we plot the distribution of the number of active latents per sample in Figure 8. We observe that BatchTopK indeed uses a wide range of active latents, activating only a single latent on some samples and activating more than 80 on others. The peak on the left of the distribution are model activations on the BOS-token. This serves as an example of the advantage of BatchTopK: when the model activations do not contain much information, BatchTopK does not activate many latents, whereas TopK would use the same number of latents regardless of the input.

---

[1] https://huggingface.co/datasets/openwebtext

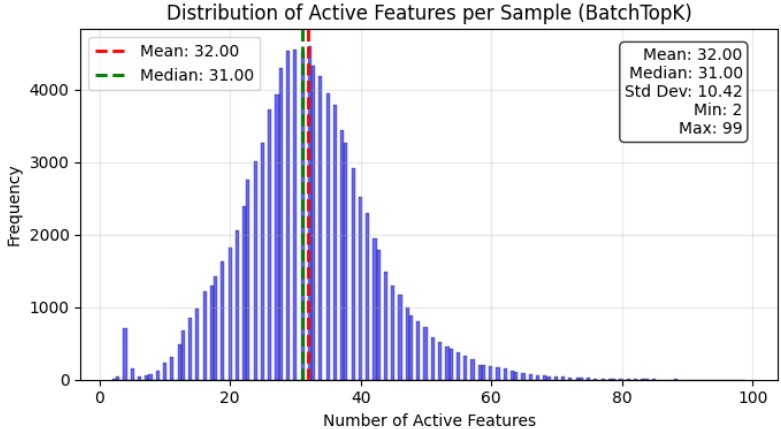

Figure 8: Distribution of the number of active latents per sample for a BatchTopK model. The peak on the left likely corresponds to BOS tokens, demonstrating BatchTopK's adaptive sparsity.

### A.4 EXAMPLE LATENTS

Figure 9 shows a histogram of the maximum decoder cosine similarity for each latent in GPT2-1536 over all latents in GPT2-768. On the right-hand-side, there is a cluster of latents with high cosine similarity.

Bricken et al. (2023) use the cosine similarity between latent activations as a measure for latent similarity. However, we use decoder cosine similarity due its lower computational cost and because it captures the latent's effect on the reconstruction. Empirically, we find a high correlation between these two metrics at values of decoder similarity relevant to our stitching experiments (see Figure 10).

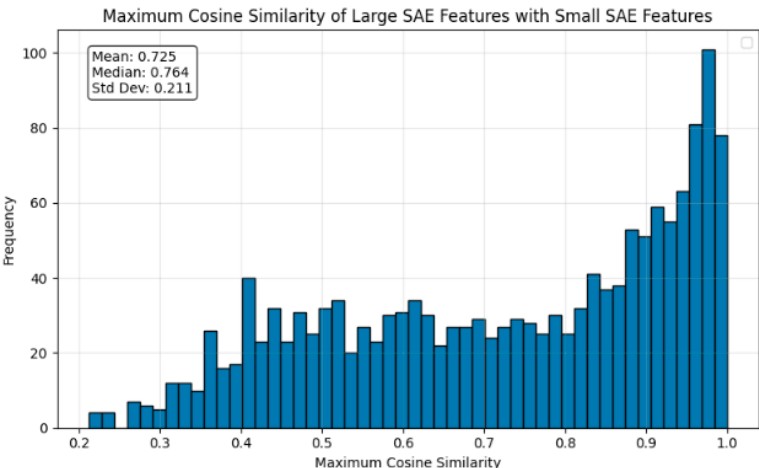

Figure 9: Distribution of maximum cosine similarities between decoder weights of latents in GPT2-1536 and GPT2-768. Many latents in the larger SAE have high similarity to latents in the smaller SAE, but there is also a long tail of novel latents.

Figure 11 and Figure 12 show feature dashboards from Neuronpedia (Lin, 2023). Feature dashboards are a popular method to visualize and interpret SAE latents, see also Bricken et al. (2023). These dashboards show tokens in input sentences that maximally activate a certain latent in green, and the tokens they boost (in blue) and suppress most (in red).

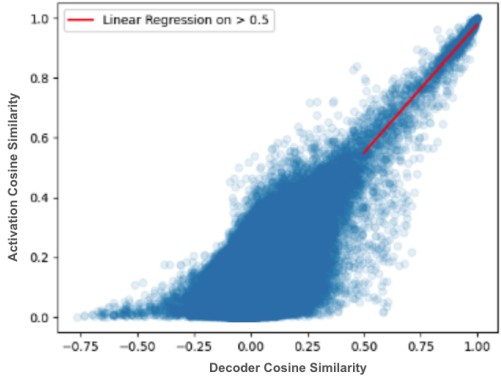

Figure 10: The activation similarity by the decoder cosine similarity on the GPT2-768 and GPT2-1536 SAEs. At high values of cosine similarity ($> 0.5$) the coefficient of determination between these is 0.87.

Figure 11 shows an example of a latent from GPT-1536 and a latent from GPT-768 that have a cosine similarity of 0.99. We see that both of these latents activate strongly on the same inputs, and boost similar logits.

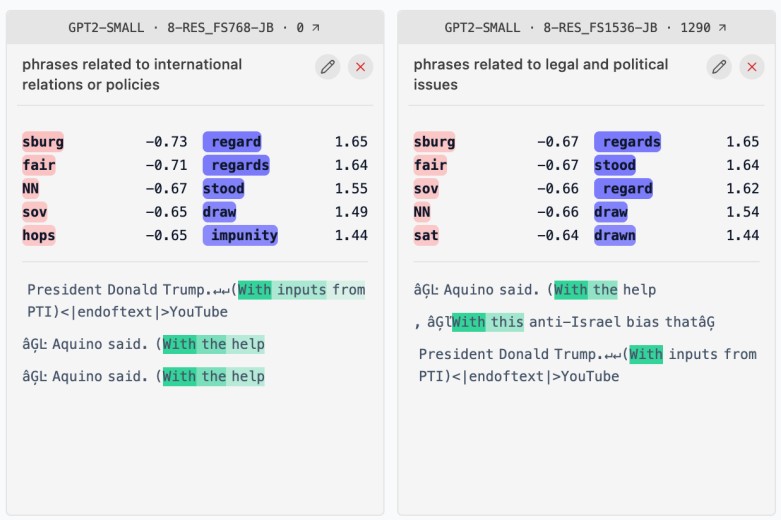

Figure 11: Neuronpedia dashboard of example latents with high cosine similarity. Neuronpedia

However, GPT2-1536 has a latent for "make sure" that has no counterpart in GPT-768. The nearest latents have a decoder cosine similarity of around 0.3, and are shown in

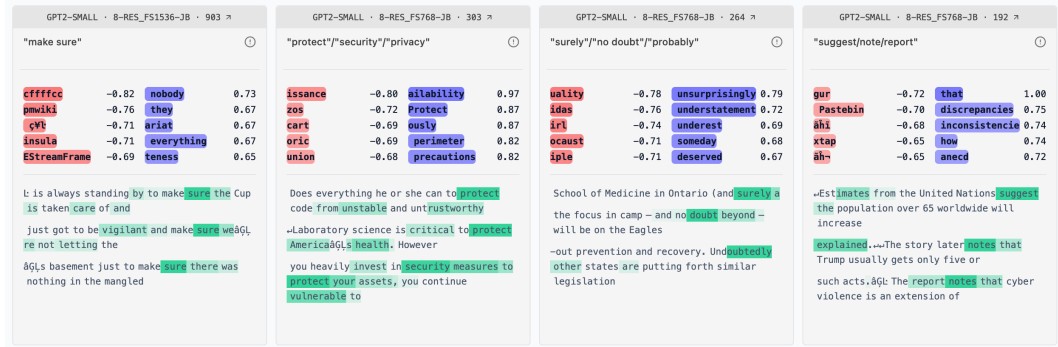

Figure 12: Example GPT2-1536 latent with no similar latent in GPT-768, with the three most similar latents shown. Neuronpedia

We evaluate the reconstruction performance of the two SAEs on inputs where this latent is active and inactive. The reconstruction performance of the smaller SAE is considerably worse on inputs where this larger SAE latent is active, compared to inputs where the latent is not active.

|  | **Latent inactive** | **Latent active** | **Difference** |
|---|---|---|---|
| GPT2-1536 | 2.225 | 2.518 | 0.293 |
| GPT2-768 | 2.703 | 3.292 | 0.589 |

Averaging this metric across all 657 latents in GPT-1536 that have low maximum cosine similarity with all latents in GPT-768, we see a similar pattern (Figure 13)

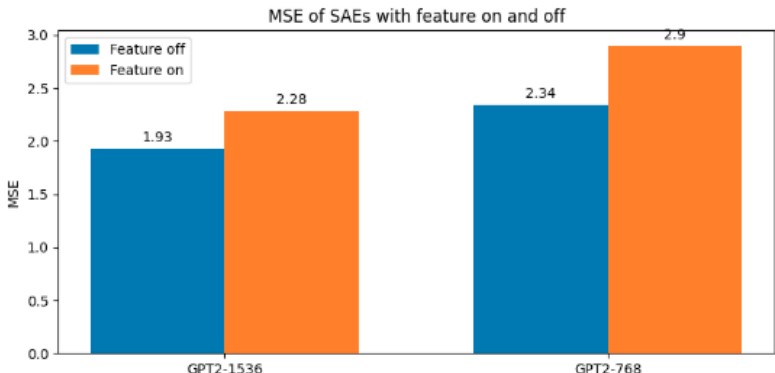

Figure 13: Reconstruction MSE of SAEs on inputs where novel latents in the larger SAE are active and inactive

## A.5 LATENT FAMILIES

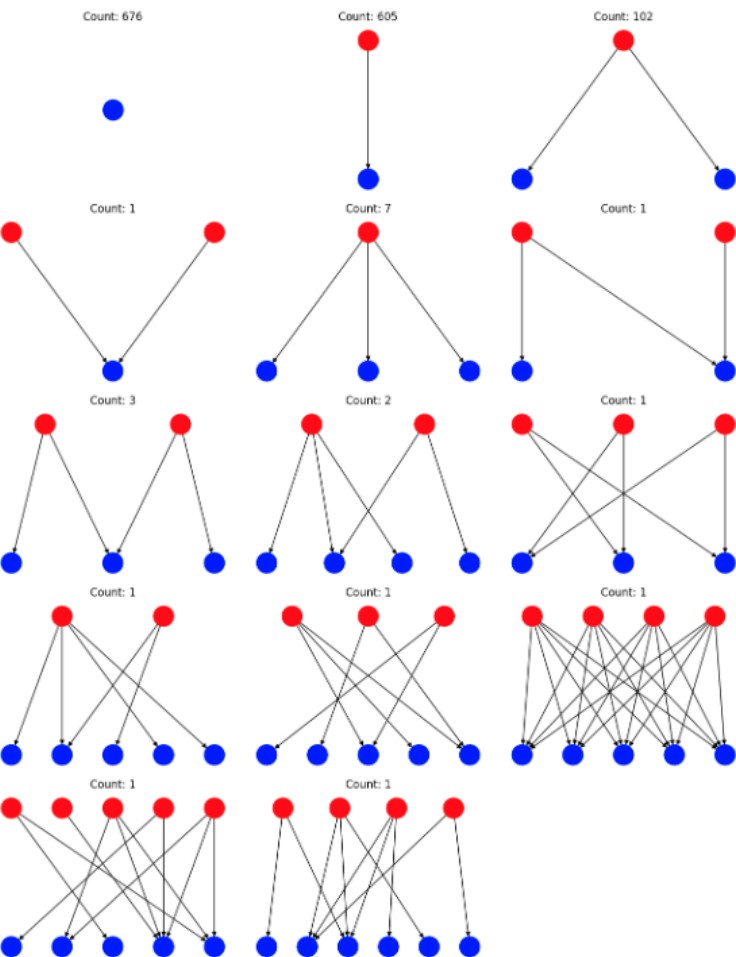

Figure 14: Connected subgraphs of the bipartite graph of latent in GPT2-768 and GPT2-1536.

## A.6 OPEN SOURCE SAE WEIGHTS

All GPT-2 Small SAEs were trained on the layer 8 residual stream, which was chosen in line with Gao et al. (2024). They were trained for 300M tokens on the OpenWebText dataset, which was processed into sequences of a maximum of 128 tokens for input into the language models. All models were trained using the Adam optimizer with a learning rate of $4 \times 10^{-4}$, $\beta_1 = 0.9$, and $\beta_2 = 0.99$. The batch size used was 4096 and all were trained with a sparsity penalty of $8 \times 10^{-5}$. The GPT-2 SAEs are available on Neuronpedia. We also use two of the Gemma Scope SAEs (Lieberum et al., 2024) trained on Gemma 2 2B (Team et al., 2024) with dictionary size 16384 and 32768. We used the TransformerLens (`https://transformerlensorg.github.io/ TransformerLens/`) implementations of GPT-2 and Gemma 2 2B. CELR is the cross entropy loss recovered from either zero or mean ablation.

Table 2: Properties of the SAEs used in this study.

| Name | Model | Dict. size | L0 | MSE | CELR Zero | CELR Mean |
|---|---|---|---|---|---|---|
| GPT2-768 | gpt2-small | 768 | 35.2 | 2.72 | 0.915 | 0.876 |
| GPT2-1536 | gpt2-small | 1536 | 39.5 | 2.22 | 0.942 | 0.915 |
| GPT2-3072 | gpt2-small | 3072 | 42.4 | 1.89 | 0.955 | 0.937 |
| GPT2-6144 | gpt2-small | 6144 | 43.8 | 1.63 | 0.965 | 0.949 |
| GPT2-12288 | gpt2-small | 12288 | 43.9 | 1.46 | 0.971 | 0.958 |
| GPT2-24576 | gpt2-small | 24576 | 42.9 | 1.33 | 0.975 | 0.963 |
| GPT2-49152 | gpt2-small | 49152 | 42.4 | 1.21 | 0.978 | 0.967 |
| GPT2-98304 | gpt2-small | 98304 | 43.9 | 1.14 | 0.980 | 0.970 |
| GemmaScope-16384 | Gemma 2 2B | 16384 | 43.4 | 1.72 | 0.983 | 0.980 |
| GemmaScope-32768 | Gemma 2 2B | 32768 | 41.5 | 1.59 | 0.985 | 0.982 |

## A.7 STITCHING EXPERIMENTS

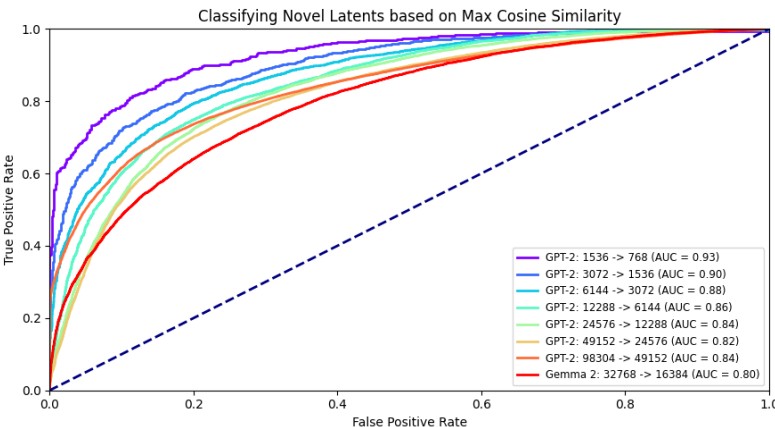

Figure 15: Based on the maximum cosine similarity it is possible to predict the direction of the effect of adding a latent to another SAE. The Receiver operating curve (ROC) is created by varying the threshold of maximum cosine similarity to classify a latent as novel latent or reconstruction latent.

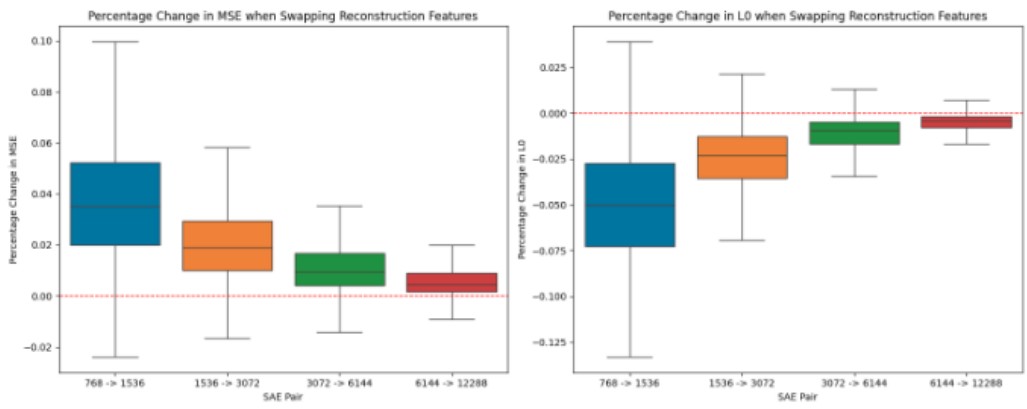

Figure 16: Effects on MSE and L0 when swapping reconstruction latents from larger SAEs to smaller ones. Swapping latent structures generally increases the MSE but almost always decreases L0. Outliers are not shown. The percentual effects per swap get smaller for larger models as the effects are distributed over more swaps.

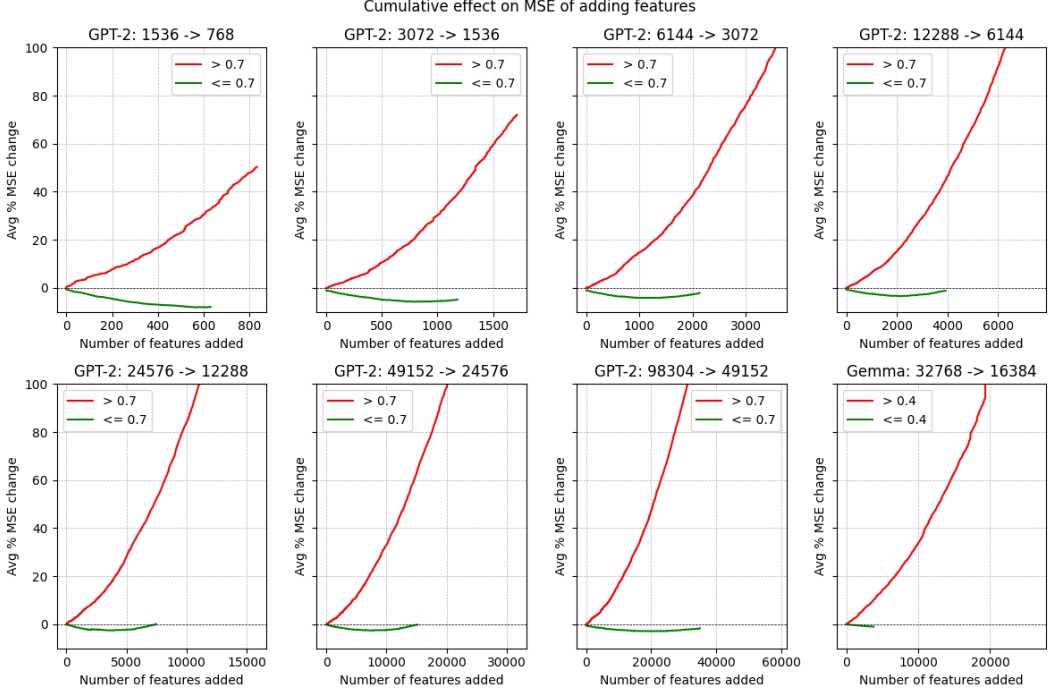

Figure 17: Percentage change of MSE of adding in latents from a larger SAE to a smaller SAE in a random order. Adding in all the latents with cosine similarity $\leq 0.7$ from GPT-1536 in GPT-768 reduces the MSE by almost 10%.

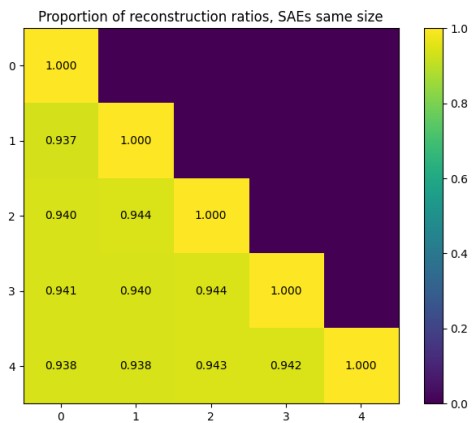
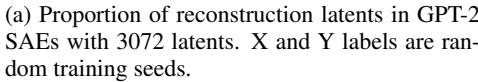

(a) Proportion of reconstruction latents in GPT-2 SAEs with 3072 latents. X and Y labels are random training seeds.

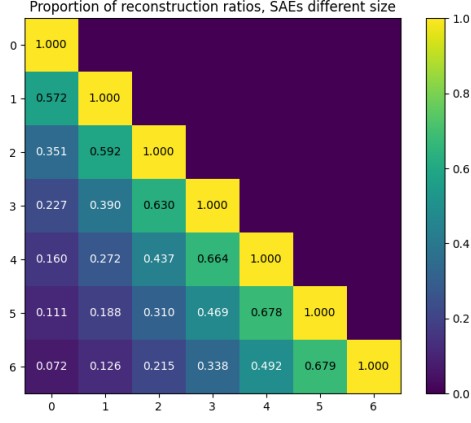

(b) Proportion of reconstruction latents in SAEs of different size. X and Y labels are exponential expansion factors, or equivalently row indices in Table 2.

Figure 18: Proportion of reconstruction latents between GPT-2 SAEs of the same size and different sizes.

We compared the proportion of reconstruction latents between SAEs of the same size and SAEs of different sizes. We found that when comparing SAEs with the same size, 94% of the latents are reconstruction latents (cosine similarity $> 0.7$). These results are displayed in Figure 18.

A concern when stitching two different SAEs is the choice of $\mathbf{b}^{dec}$. However, in practice we find that the $\mathbf{b}^{dec}$s of SAEs trained on the same latents are very similar (minimum cosine similarity of 0.9970, differing by less than 0.1% in magnitude). Figure 19 shows that the decoder biases can be

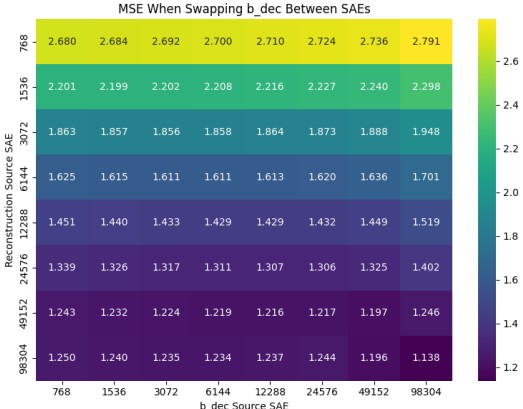

Figure 19: Reconstruction performance (MSE) of SAEs of different sizes when their $\mathbf{b}^{dec}$s are replaced with the $\mathbf{b}^{dec}$s of SAEs with a different dictionary size.

exchanged in the stitching process with negligible impact on the reconstruction performance of an SAE.

### A.7.1 STITCHING IN GEMMA SCOPE SAES

In order to validate that the SAE stitching results are not an artifact of GPT-2 small or the SAEs that we trained, we applied the same methods to the open-source Gemma Scope SAEs. In particular, we compared two SAEs trained on the residual stream of layer 12 of Gemma 2 2B. The first SAE has dictionary size 16384 (average L0 41) and the second SAE has dictionary size 32768.

We find a lower threshold for distinguishing novel features from reconstruction features (0.4). Using this threshold, we can also smoothly interpolate between SAEs of different sizes trained on Gemma-2-2B.

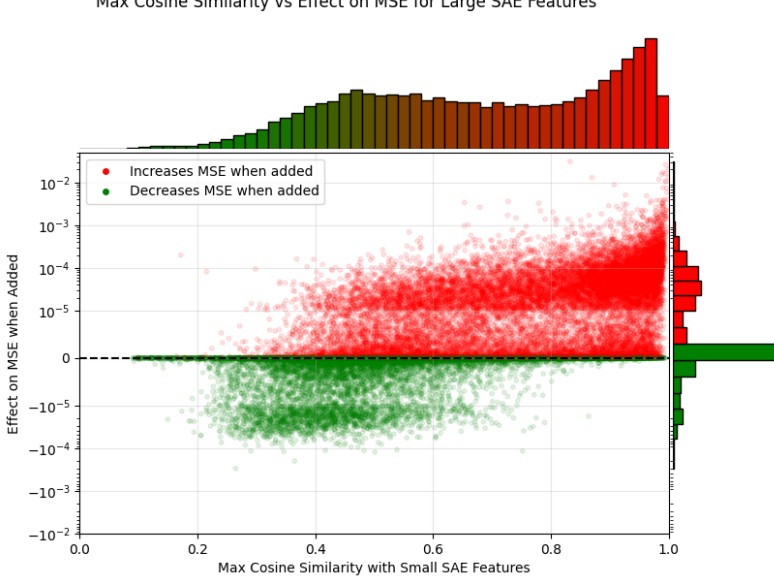

Figure 20: Change in MSE when adding each feature from GemmaScope-32k to GemmaScope-16k, plotted against the maximum cosine similarity of that feature to any feature in GemmaScope-16k. Features with cosine similarity less than 0.4 tend to improve MSE, while more redundant features hurt performance.

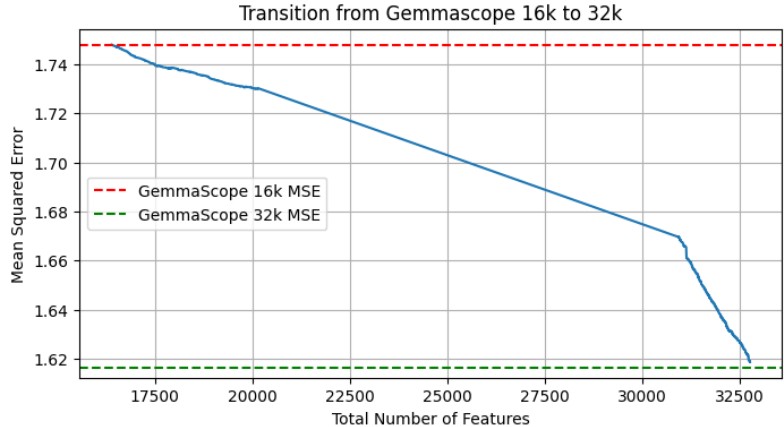

Figure 21: By first adding in novel latents from Gemma Scope 32k to Gemma Scope 16k and replacing the remaining latents with their similar latents in the larger SAE, we interpolate between the two SAE siezs.

## A.8 MetaSAE Additional Figures

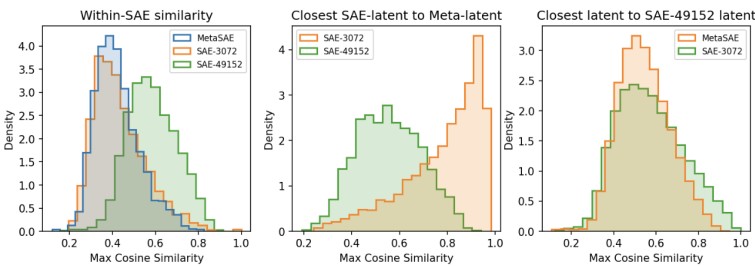

Figure 22: Cosine similarity between SAE latents and meta-SAE latents. Note the high maximum cosine similarity between latents from a meta-SAE with 2304 latents, and a standard SAE with 3072 latents.

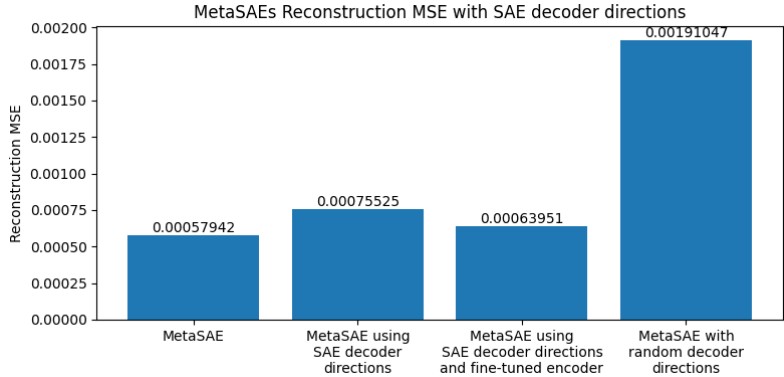

Figure 23: Change in reconstruction performance of a meta-SAE when its decoder directions are replaced with the most similar decoder direction from an SAE with a similar dictionary size.

## A.9 INTERPRETABILITY EXPERIMENTS

In this paper, we demonstrate that larger SAEs may learn more narrow, composed concepts in order to improve sparsity rather than just learning concepts that are missing in smaller SAEs. Here, we provide some experimental results on sparse probing Gao et al. (2024) and concept removal benchmarks (Karvonen et al., 2024).

### A.9.1 SPARSE PROBING

Similarly to Gao et al. (2024), we use sparse probes to evaluate the presence of known ground-truth features in our SAEs. If we expect a specific feature to be discovered by an SAE, then a metric for autoencoder quality is simply checking whether these features are present as latents. We do this by training a 1-dimensional logistic probe on the activations of the SAE to predict the presence of the feature. We use the benchmark datasets included in (Karvonen et al., 2024) in our evaluation. These cover a range of domains, for example predicting the sentiment of Amazon reviews or predicting the language of the text from the SAE activations.

In our experiments we use a probe that uses only a single SAE latent in its prediction. The results of these experiments are visualized in Figure 24. They show that the relationship between the size of the SAE and the evaluation accuracy is complex and dataset dependent.

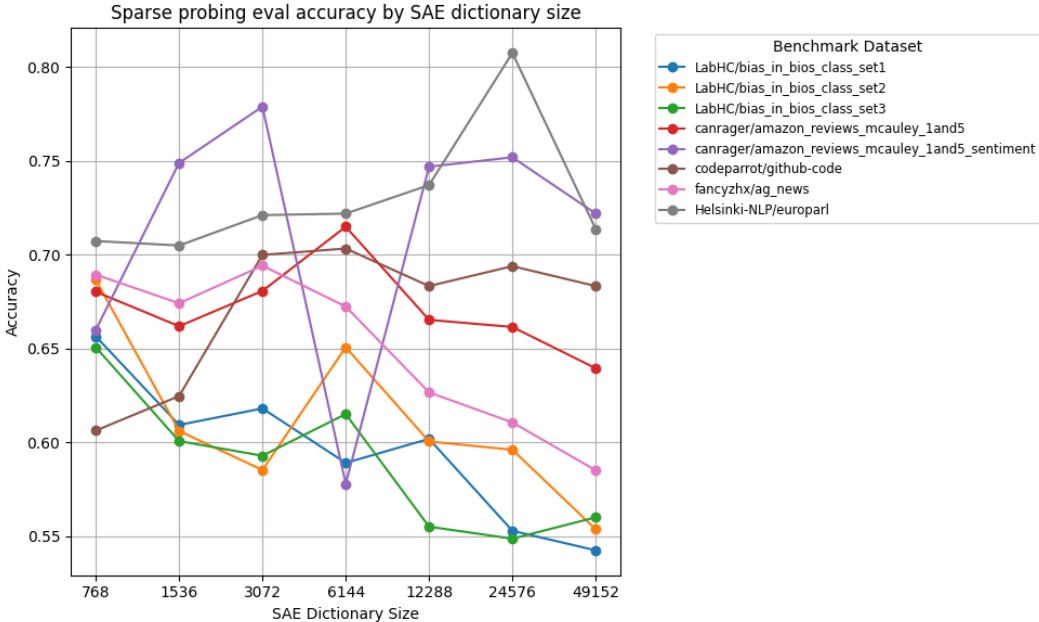

Figure 24: Sparse probing evaluation accuracy by GPT-2 SAE dictionary size across 8 benchmark datasets, with a sparse probe using the top latent.

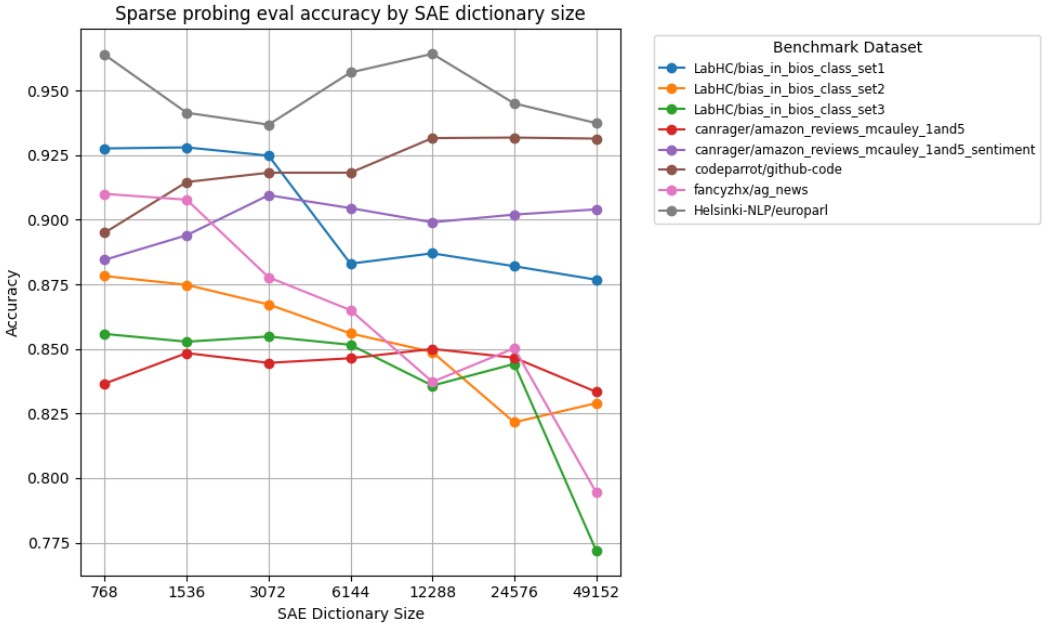

Figure 25: Sparse probing evaluation accuracy by GPT-2 SAE dictionary size across 8 benchmark datasets, with a sparse probe using the top 50 latents.

### A.9.2 CONCEPT REMOVAL

In the SHIFT method (Marks et al., 2024), a human evaluator debiases a classifier by ablating SAE latents. (Karvonen et al., 2024) operationalizes SHIFT into an evaluation for SAE quality, Targeted Probe Perturbation, by training probes on model activations and measuring the effect of ablating sets of SAE latents on the probe accuracy. Ablating a disentangled set of latents should have an isolated causal effect on one class probe, while leaving other class probes unaffected.

They consider a dataset with $M$ classes. For each class with index $i = 1, \ldots, M$ they select the set $L_i$ of the most relevant SAE latents. To select those latents, binary probes are trained detecting the concept $c$ from model activations. The attribution score of latent with index $l$ on concept $c$ is given by

$$I(L, c) = (L_{pos} - L_{neg})(\mathbf{d}_l \cdot \mathbf{P}) \tag{11}$$

where $L$ denotes the batch of activations of latent $l$, $\mathbf{d}_l$ denotes the SAE decoder vector corresponding to $l$, $\mathbf{P}$ is the weight matrix of the binary probe, $L_{pos}$ is the mean activation of the latent for inputs of the targeted concept, and $L_{neg}$ is the mean activation for inputs unrelated to the concept. The latents with the highest attribution score are those select as most relevant.

For each concept $c_i$, they partition the dataset into samples of the targeted concept and a random mix of all other labels. They define the model with probe corresponding to class $c_j$ with $j = 1, \ldots, M$ as a linear classifier $C_j$. Further, $C_{i,j}$ denotes a classifier for $c_j$ where latents $L_i$ are ablated. Then, they iteratively evaluate the accuracy $A_{i,j}$ of all linear classifiers $C_{i,j}$ on the dataset partitioned for the corresponding class $c_j$. The targeted probe perturbation score

$$S_{\text{TPP}} = \text{mean}_{(i=j)}(A_{i,j}) - \text{mean}_{(i \neq j)}(A_{i,j}) \tag{12}$$

represents the effectiveness of causally isolating a single probe. Ablating a disentangled set of features should only show a significant accuracy decrease if $i = j$, namely if the latents selected for class $i$ are ablated in the classifier of the same class $i$, and remain constant if $i \neq j$

The results of this evaluation by SAE dictionary size is displayed in Figure 26. Whilst here there is a general downward trend, the accuracy is not monotonically decreasing with SAE size.

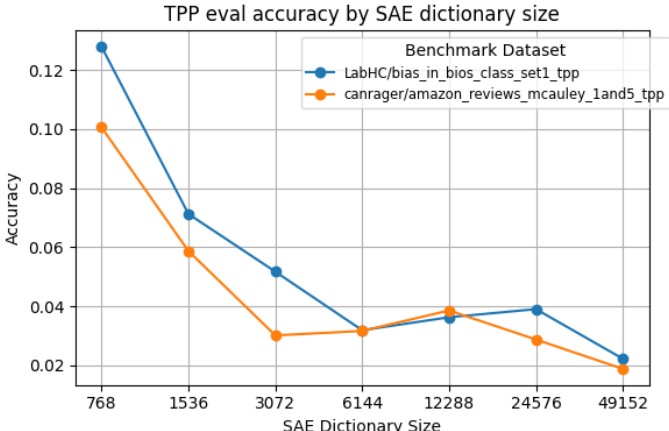

Figure 26: TPP evaluation accuracy by GPT-2 SAE dictionary size across 2 benchmark datasets, ablating up to 50 latents.

