# OpenReview forum: "Sparse Autoencoders Do Not Find Canonical Units of Analysis"
_ICLR.cc/2025/Conference — ICLR 2025 Poster_

### Official Review · Reviewer_9gei · 2024-10-31

**Soundness:** 3
**Presentation:** 2
**Contribution:** 2
**Rating:** 6
**Confidence:** 3

**Summary:**

This research concerns SAE, an assessment tool specialized for GPT-type models, espeically those used as LLM.
SAE is applied on the activation of LLM to infer the captured information, and at least in this paper, its wellness is
evaluated based on the MSE of the reconstruction.
The novelties of of this research are as follows.
1. It proposes SAE stitching, a tool to compare large SAE against Small SAE.  With SAE stitching, one can use the
change in MSE to assess how the larger SAE's weight direction relates to the smaller SAE's weight direction (intersection of notions),
as well as the uniquness of the notion captured by the larger SAE.
2.  It proposes Meta SAE, which applies still another SAE on the top of an already established SAE. It is used to obtain the
monosemantic latent.
3.  It proposes BatchTopK, a variant of MetaSAE which achieves SOTA in terms of architecture.
Base on the observations made by these claimed novel technique set, the paper advocates the need to carefully choose the size of the
SAE as well as the need to compare SAEs of different sizes for more semantically meaningful analysis.

**Strengths:**

- This research provides thorough and solid experiments that are in alignment with
those of the original SAE paper.
- This research furthers the understanding of SAE in application to LLM; considering that SAE is considered an important probing unit in the understanding of how LLM processes the information, this research answers an important question regarding (1) how the probing unit deomposes the LLM activations and (2) how its size and design affects the analysis.

**Weaknesses:**

- The reviewer is a little unsatisfied with how the research is positioned. In the introduction and in the abstract, the paper is presented as if the research topic pertains to an analysis of a decomposition of the activations of a "general" neural network into features, while in fact the SAE introduced by [Bricken et al] and [Cunningham et al] is a probing tool for a specific 'organism' that is LLM.  The term 'language model' is mentioned only at the part 3 of the contribution, leaving an impression that the paper investigates a very general feature analysis that is 'also' applicable to LLM.  The research also experiments with LLM exclusively as well.
While the keyword SAE may automatically link to the behavioral study of LLM in the minds the the readers that are actively involved in LLM behavioral research, the reviewer feels that it shall be clearly stated/emphasized in both the introduction and the abstract that this research is about LLM (which is merely one genre of ML research).  Otherwise, the reviewer believes the applications other than LLM shall be presented in the paper.

- Another concern is that, while the research clearly furthers the understanding of the SAE, an important probing tool for LLM, the research does not relate the claimed novelty to the SAE's probing "capability". For example, [Cunningham et al] quantifies the SAE's ability to localize a specific model behavior in Indirect object Identification (IoI) task, thereby evaluating the goodness of the probing conducted by SAE.  Meanwhile, the paper evaluates the goodness of SAE with the reconstruction error. While the reviewer values the thoroughness of independent experiments, the reviewer also feels that the authors did not intend their study to be interpreted as an investigation of a probing tool for the sake of probing tool itself.  The reviewer feels that the research can be justified if the authors can add analysis of how their new toolset can be used to better the probing capability of SAE in terms of IoI, for example, or to actually better uncover a features that are causally responsible for counterfactual behavior in LLM.

**Questions:**

1,  Figure 4, it is said that "Features with cosine similarity less than 0.7 tend to improve MSE".  Do you mean 0.6 or less?  In the similar regard, thsi rendency is much less clear in Figure 18.  Would you please make comments regarding GemmaScope32k? It is said that " Using this threshold, we find that our feature stitching methods work on these SAEs as well." Can you elaborate on what is meant by "well" in this context?

2,  What are "active latents"?  The reviewer presumes that these are the latents that are not "zero-ed out" by the training with the sparsity constraint, but it would help to clarify its precise mathematical meaning.

3, In Figure 6 and experiments regarding this figure, MetaSAE with N latents are evaluated.  Now, the MetaSAE is presented as the application of SAE on the latents of SAE.   Would you please clarify "on what SAEs" the MetaSAE were applied in these experiments? For example, from SAE of what size "the Meta SAE with 2304 latents" was obtained?

---

> ### Author Response · Authors · 2024-11-20
> **Response to Reviewer 9gei**
>
> Thank you for your thoughtful review and helpful suggestions. In particular, we were happy with your assessment that our work “provides thorough and solid experiments” and “answers an important question”. We appreciate your feedback on the positioning of the paper within the literature, and have updated the manuscript to be more precise about the modality of the experiments, and added experiments to clarify the relevance of our results to interpretability. We address your main points and questions below. If you have further comments or questions we are keen to respond to them, and otherwise appreciate an increase in your support for our paper.
>
> > In the introduction and in the abstract, the paper is presented as if the research topic pertains to an analysis of a decomposition of the activations of a "general" neural network into features, while in fact the SAE introduced by [Bricken et al] and [Cunningham et al] is a probing tool for a specific 'organism' that is LLM.
>
> SAEs have been used on a range of modalities, including image and audio models. However, you are correct that our work specifically advances the understanding of SAEs as interpretability tools for LLMs. We have updated the abstract and introduction to specify this.
>
> > The research does not relate the claimed novelty to the SAE's probing "capability"
>
> Please see the top-level comment.
>
> > Figure 4, it is said that "Features with cosine similarity less than 0.7 tend to improve MSE". Do you mean 0.6 or less? In the similar regard, this tendency is much less clear in Figure 18.
>
> We found that different model architectures and training regimes lead to different optimal thresholds (0.7 for GPT-2, 0.4 for Gemma). By "work well" we mean that using these thresholds allows qualitatively similar results (such as the interpolation between SAE sizes), though the exact patterns differ between models. We have updated the text to clarify this. Note that by changing the threshold, we can adapt the trade-off between false positives and false negatives, as shown in the ROC plot in Figure 16.
>
> > What are "active latents"?
>
> We have added a definition to the Glossary of Terms (Appendix A.1) - these are latents with non-zero activation values after applying the sparsity-inducing activation function.
>
> > Would you please clarify "on what SAEs" the MetaSAE were applied in these experiments? For example, from SAE of what size "the Meta SAE with 2304 latents" was obtained?
>
> The meta-SAE with 2304 latents was trained on the decoder directions of the GPT-2 SAE with 49152 latents. These experimental details can be found in line 403 in the paper.

---

> ### Author Response · Authors · 2024-11-25
>
> Before this phase of the rebuttal period ends, we wanted to ask the reviewer whether we have addressed your concerns with our work?

---

> > ### Comment · Reviewer_9gei · 2024-11-25
> > **Thank you very much for the feedback!!!**
> >
> > I am sorry in the delayed response, and thank you very much for addressing most of my concerns.
> > Thank you also for updating the abstract as well. My remaining concern is just the relation between the claimed novelty to the SAE's probing "capability" in actual downstream tasks.
> > If my understanding of your response comment is correct,  I believe that the said "top-level comment" in response to my concern is
> >
> > >  To reinforce this argument, we have added two sets of interpretability experiments to Appendix A.9, that show the complex relationship between the usefulness of latents for down-stream interpretability tasks and size of the dictionary.
> >
> > In this regard, I was sweeping through the Appendix section, and I am postulating that the two sets of "interpretability experiments" in mention are the numerical report of A.4 regarding the similarities between Large and small SAE features.  Would you please be more specific as to which "additional experiments"  you are referring to, and which downstream task (such as IoI) is being discussed there?

---

> > > ### Author Response · Authors · 2024-11-25
> > > **Clarification on new interpretability experiments**
> > >
> > > Thank you for clarifying your concern about the additional interpretability experiments. The two interpretability experiments we are referring to can be found in Appendix A.9 of the updated version. The first experiment (in Appendix A.9.1) shows how the latents of SAEs of different sizes can be used as linear probes for multiple classification tasks, such as classifying the occupation of a person based on online biographies, predicting the sentiment of amazon reviews, and the category of news articles (e.g. world, sports, business).
> > >
> > > The second experiment (in Appendix A.9.2) scores how well ablating latents from SAEs of different sizes removes information from the model activations. Here we use the same tasks (i.e. the sentiment of amazon reviews), but score how well SAE latents can be used to remove the class information without impacting other information present in the activations.
> > >
> > > We are happy to see that we have addressed most of your concerns. Do these experiments resolve your remaining concern regarding the probing capability in downstream tasks? We would greatly appreciate it if you would consider raising your score for the paper.

---

> > > > ### Comment · Reviewer_9gei · 2024-11-26
> > > > **Thank you very much for the followup!**
> > > >
> > > > Thank you for addressing my concern, I am more comfortable with the claim now.
> > > > While I am positive about the score however, there seems to be one last bit; I am worried about the citation of [Upcoming Work] that are being referred either as an evaluation method in the said appendix or as a benchmark in the main manuscript. Would you please clarify what is [Upcoming Work] ?

---

> > > > > ### Author Response · Authors · 2024-11-26
> > > > >
> > > > > Thank you for your feedback, we’re glad we were able to improve your confidence in the research!
> > > > >
> > > > > > I am worried about the citation of [Upcoming Work] that are being referred either as an evaluation method in the said appendix or as a benchmark in the main manuscript. Would you please clarify what is [Upcoming Work] ?
> > > > >
> > > > > Thanks for raising this concern. [Upcoming Work] is a SAE benchmark being developed by researchers we know personally, who were kind enough to let us build on their work before the official publication date, intended for early December. This work will be published well before the decision deadline, so the current awkward state of the manuscript will be fixed in the camera ready with a clear citation to their work. In order to avoid confusion, or to claim any undue credit for their methodology, we tried to make it clear that this was upcoming work. We have added a footnote to Appendix A.9 clarifying this.
> > > > >
> > > > > We hope this has addressed your remaining concern, but please feel welcome to ask for any further clarifications during the next stage of the review process. If you are now satisfied with the manuscript, we would politely ask that you increase your support for the paper.

---

> ### Comment · Reviewer_9gei · 2024-11-28
> **Thank you for the clarification**
>
> Thank you for clarifying the [Upcoming Work], and I am very sympathetic to the situation.
> While so, I have never seen this case in my moderately long research career so far, particularly the case in which the
> reference is being made to a work that has not been even unofficially published (e.g. ArXiv) so that no one can verify its validity.
> While I do feel the sympathy, please understand that I must also play my part in conducting a fair review and that the problem addressed in this part of the additional experiment is very important in my opinion.
> By any chance, would it be possible to point out any preceding example in which something like this has been done before, so that I feel more confident in raising the score?  Meanwhile, I will try to look for a such example myself within my capability.

---

> > ### Author Response · Authors · 2024-11-28
> >
> > We understand your concern, and really appreciate your commitment to conducting a fair review of our work. We haven’t managed to find other examples of this, as it seems quite hard to search for. In order to help you validate the reference, we have been granted permission to unofficially publish the paper and code on Github. We hope that this is an acceptable temporary solution for you in conducting your review, and we apologise for the additional complexity this has created for you. We have updated the citation to reflect this, and the paper and code are available here: https://github.com/anonymous664422/sae_bench

---

> ### Comment · Reviewer_9gei · 2024-11-28
> **Thank you very much for the resolving the concern**
>
> Thank you very much for trying out the best to resolve the concern. It was very important to me that the experiment is being done on the benchmark that are available for all reviewers (including myself) to see.  I am raising the score. Please notify to all reviewers that this modification on your manuscript has been done (including the github link).

---

### Official Review · Reviewer_HcSs · 2024-10-31

**Soundness:** 3
**Presentation:** 3
**Contribution:** 2
**Rating:** 8
**Confidence:** 3

**Summary:**

In this work, the authors introduce two new methods for analyzing latents in SAEs. The first, SAE stitching, enables comparison of latents across SAEs of different sizes by categorizing latents in larger SAEs as either novel or reconstruction latents found in smaller models. The second, Meta-SAE, decomposes decoder directions into interpretable meta-latents They also propose the BatchTopK SAE, a variant of the TopK SAE that forces a fixed average sparsity. By applying these methods, they obtained empirical results that suggest that SAEs are unlikely to learn canonical sets of units.

**Strengths:**

- The motivation and methods are explained clearly and intuitively, with helpful examples.
- The authors contextualize their approach by discussing relevant state-of-the-art methods.
- Several experiments are conducted to assess the methods' performance, with comparisons to state-of-the-art baselines and detailed experimental information.
- An interactive dashboard is included to explore the latents learned with meta-SAEs.

**Weaknesses:**

- Including a discussion on the potential limitations of the proposed approaches would be valuable.
- It would also be helpful to have an expanded discussion on assessing the quality of the representations and adapting the dimensionality of the SAE to suit the requirements of different analyses.

**Questions:**

- Is there any particular reason to omit the bias term of the decoder 1 in equation 5?
- How does the stitching SAE handle shared reconstruction latents during the swap process? Since some reconstruction latents in larger SAEs are composites, they may contain information from multiple smaller SAE latents. For example, in the case of colored shapes, how would the model swap the "square" or "blue" latent if these are entangled in composite latents in the larger SAE? Would they need to be swapped simultaneously?
- Would it be possible to introduce some form of supervision in meta-SAEs? Do you have any intuition as to whether this might be beneficial, considering that the ultimate goal is to interpret model activations? Following a concept bottleneck approach, one could directly associate human-interpretable meta-latents with the representations learned by larger SAEs (although I assume the main limitation is defining labels/concepts for the large dictionary sizes considered).
- For BatchTopK, have the authors examined how varying values of k impact the semantics of the learned latents? Do the "concepts" learned under a stronger sparsity constraint become more abstract (as they need to explain a given activation with fewer latents) for a fixed dictionary size?
- Have the authors extracted any intuition on how should the dictionary size be adjusted to tailor the SAE to the requirements of an specific analysis?

---

> ### Author Response · Authors · 2024-11-20
> **Response to Reviewer HcSs**
>
> Thank you for your positive review and thoughtful questions. We appreciate your assessment that our “motivation and methods are explained clearly and intuitively, with helpful examples” and that we provide “detailed experimental information”. We address your main points and questions below:
>
> > Including a discussion on the potential limitations of the proposed approaches would be valuable.
>
> We have expanded the conclusion section, highlighting the limitations of our work.
>
> > Is there any particular reason to omit the bias term of the decoder 1 in equation 5?
>
> For a given base language model, the decoder biases of all the SAEs are very similar. For GPT-2, the minimum cosine similarity between any two pairs of SAE decoder biases is 0.9970, and the magnitude of the vectors differs by less than 0.01%, so we arbitrarily use b^{dec}_0. We've updated the text around the equation to clarify this.
>
> > How does the stitching SAE handle shared reconstruction latents during the swap process? For example, in the case of colored shapes, how would the model swap the "square" or "blue" latent if these are entangled in composite latents in the larger SAE?
>
> In case of one-to-many, many-to-one, or many-to-many relationships between reconstruction features in two SAEs, they are indeed swapped simultaneously. Please see Appendix A.5 for some examples of the sub-graphs of swaps that were performed on the two smallest GPT-2 SAEs. In the case of the “blue square”-example, this would result in a sub-graph containing all the latents in both SAEs.
>
> > Would it be possible to introduce some form of supervision in meta-SAEs?
>
> We believe the unsupervised nature of both SAEs and meta-SAEs is actually a key advantage, as it allows us to discover what compositional structure the model naturally represents rather than imposing our assumptions about what concepts should exist. We agree that future work with supervised datasets could be valuable for validating (meta-)SAE performance and understanding how discovered decompositions align with human concepts.
>
> > For BatchTopK, have the authors examined how varying values of k impact the semantics of the learned latents? Do the "concepts" learned under a stronger sparsity constraint become more abstract (as they need to explain a given activation with fewer latents) for a fixed dictionary size?
>
> Our paper primarily focuses on the effect of dictionary size on the abstractness of features, rather than the role of sparsity (k). However, we agree that analyzing the relationship between sparsity and semantic granularity would be valuable future work, both in BatchTopK SAEs and other SAE architectures.
>
> > Have the authors extracted any intuition on how should the dictionary size be adjusted to tailor the SAE to the requirements of an specific analysis?
>
> We have addressed this in the top-level comment.

---

> > ### Comment · Reviewer_HcSs · 2024-11-22
> > **Response to Authors**
> >
> > Thank you for your response! I believe that this paper makes a positive contribution to our understanding of SAEs, which could ultimately inspire new research directions to address the main limitations identified (i.e. SAEs not identifying canonical units of analysis and the challenge of determining the appropriate dictionary size for a given task). I also appreciate the improvements made in the revised version - it is now easier to read, more self-contained thanks to the glossary, and the experimental section is more comprehensive. I am maintaining my rating, as I believe this is a good paper worth sharing with the community.

---

### Official Review · Reviewer_pzvM · 2024-11-02

**Soundness:** 3
**Presentation:** 2
**Contribution:** 2
**Rating:** 6
**Confidence:** 4

**Summary:**

The paper discusses the use of sparse autoencoders (SAEs) to examine how activations in a transformer relate to concepts in natural language. The authors examine the relationship between small SAEs and larger SAEs (a larger number of latent states) to assess whether larger SAEs contain more fine-grained information. Additionally the authors introduce a sparsity objective that enforces sparsity across a batch of feature vectors, rather than at for individual feature vectors.

**Strengths:**

The paper attempts to understand how activations deep within a transformer might correspond to higher level natural language concepts. This is an interesting topic. The conclusion is also interesting, namely that sparse autoencoders may not form a complete set of explanations, with larger SAEs potentially containing more fine-grained information.

**Weaknesses:**

The paper is quite presumptive about readers knowledge of the topic, lacking in clear explanations in places. Whilst the paper does a reasonable job of explaining sparse encoders, it doesn't explain how these are used to find actual "inputs" that activate particular features. This isn't explained in the paper and I had to read the cited papers to understand this. There is some description in section 5.1 (page 8) but this is too late for the reader unfamiliar with this area to understand the paper.

I also found it quite hard to follow the authors reasoning in places and it's not clear to me why stitching might provide insight.  Overall, the methods used in the paper are rather straightforward and I would therefore have hoped for a very clear presentation to make up for a lack of technical contributions. The Meta SAE isn't very well explained and the BatchTopK is idea is simple but now well motivated and it's unclear to me how this connects to the rest of the paper.

Try to be consistent with spelling rather than mix eg color, colour. Also opening parentheses are incorrectly used in places (see lines 369 onwards).

I do think there is something potentially insightful in this work, but the paper needs to be more clearly presented.

**Questions:**

*** page 4

I don't really agree with the phrase "circuit". To me this suggests some end to end explanation, whereas the reality is that a small part of a transformer is being examined (the activations in a single layer).

I can't find any information on how the SAE is trained. What is the optimiser and how are the hyperparameters (eg \lambda) set?

*** page 5

Why choose layer 8 of GPT2?

Equation 5 has an asymmetry. Why is the bias from decoder 0 used, rather than decoder 1, or a combination of the two?

The motivation for stitching isn't clear to me. The features and decoders learned are optimised for each SAE separately. Why would combining them in a suboptimal way mean anything? Why not fix the features f0, f1 from each encoder but then learn optimal decoder weights W_{01} for the combination of these feature (this is a simple quadratic optimisation problem). Similarly, in the subsequent discussion, I don't follow why an increase or decrease in MSE is of any significance.

If one wishes to understand whether latents in larger SAEs are finer grained versions of latents in smaller SAEs, would it not be feasible to look at a sentence that coactivates two features, one feature from the smaller SAE and one from the larger SAE?

In figure 5 it's not clear to me what the average L0 means -- what is being averaged over here?

As far as I understand the SAE objective is non-convex, meaning that there is no guarantee of finding the global optimum. It's therefore also quite possible that different SAEs are simply finding different latents simply because of finding different local minima.

Isn't it clear that (local optima aside) a larger SAE will always find features that are missed by smaller SAE? I'm not sure I follow the argument from line 355 onwards.

*** page 8

Please add more clarity around treating the latents W_i^{dec} (why is W in boldface)? W_i^{dec} is a scalar quantity, the ith component of the d-dimensional vector W^{dec}. What does it mean to take scalars "training data for our meta-SAE"? The directions W don't convey information about which directions are simultaneously activated. I would have thought it would make more sense to treat the feature vectors f(x) as the entities for learning a meta SAE.


*** page 9

The BatchTopK function is fully defined. In line 465 it states that the function selects the top K activations, suggesting that this is a mapping from activations to indices. I suspect that the authors mean that the function should return zero for all activations that are not in the top K highest positive values, and is the identity function otherwise.

The introduced batch method is used only during training, with a different non-linearity used during "inference". This seems quite strange and it's not clear to me how to justify this.  A potential issue not mentioned is that it's quite possible that the batch approach means that some input sequences will have entirely zero activation.

It's hardly surprising that the BatchTopK SAE has lower SAE than TopK SAE since BatchTopSAE imposes fewer constraints on the objective. I'm not sure why this would be seen as "outperformance".

I'm also unclear as to why BatchTopK SAE is being discussed. Is this method used in all the previous experiments in the paper, or is this a separate piece of work orthogonal to the other contributions of the paper?

*** page 10

The conclusion "These findings suggest that there is no single SAE width at which it learns a unique and complete dictionary of atomic features that can be used to explain the behaviour of the model." is interesting and (perhaps) not surprising.


*** supplementary material

Figure 11 isn't well explained. Please explain what is being shown here.

---

> ### Author Response · Authors · 2024-11-20
> **Response to Reviewer pzvM**
>
> Thank you for your extensive review! We appreciate your feedback in helping us make this paper more appealing and clear to readers from the broader community. We have run some additional experiments, and address your questions and critiques below. If you have any further questions or feedback we are keen to hear them, and otherwise we would appreciate you to increase your support for our paper.
>
> > I don't really agree with the phrase "circuit". To me this suggests some end to end explanation, whereas the reality is that a small part of a transformer is being examined (the activations in a single layer).
>
> We use the term "circuit" only in the related work section where we reference papers that examine activations across multiple layers (e.g., Marks et al.'s "Sparse Feature Circuits"). While our work focuses on single-layer analysis, we maintain this terminology when discussing prior work to be consistent with the established literature.
>
> > I can't find any information on how the SAE is trained. What is the optimiser and how are the hyperparameters (eg \lambda) set?
>
> Thanks for pointing this out. We have added the training details for the GPT-2 SAEs to Appendix A.6 The Gemma-2 SAEs are described in Lieberum et al.
>
> > Why choose layer 8 of GPT2?
>
> We followed the lead of Gao et al. (2024) in choosing the layer 8 residual stream. This clarification has been added to Appendix A.6 along with the training details.
>
> > Equation 5 has an asymmetry. Why is the bias from decoder 0 used, rather than decoder 1, or a combination of the two?
>
> For a given base language model, the decoder biases of all the SAEs are very similar. For GPT-2, the minimum cosine similarity between any two pairs of SAE decoder biases is 0.9970, and the magnitude of the vectors differs by less than 0.01%. We've updated the text around the equation to clarify this.
>
> > The motivation for stitching isn't clear to me. The features and decoders learned are optimised for each SAE separately. Why would combining them in a suboptimal way mean anything? Why not fix the features f0, f1 from each encoder but then learn optimal decoder weights W_{01} for the combination of these feature (this is a simple quadratic optimisation problem). Similarly, in the subsequent discussion, I don't follow why an increase or decrease in MSE is of any significance.
>
> While learning optimal decoder weights is possible, our goal is to understand relationships between features learned by SAEs of different sizes, not to optimize their combination. By keeping original decoder weights, we can directly measure whether larger SAE features provide novel information or reconstruct information already captured by smaller SAEs. The MSE changes directly indicate whether larger SAEs learn novel features or just different representations of the same information.
>
> > If one wishes to understand whether latents in larger SAEs are finer grained versions of latents in smaller SAEs, would it not be feasible to look at a sentence that coactivates two features, one feature from the smaller SAE and one from the larger SAE?
>
> From a single sentence it would be hard to distinguish such a relationship between features in SAEs of different sizes as many features from each SAE are likely to be active on that sample. Bricken et al. (2023) use coactivation as a measure of latent similarity, however we empirically found a high correlation between this metric and the decoder similarity metric (see Appendix A.4 Figure 11), and decoder similarity has lower computational cost. Furthermore, decoder directions define the effect of an active latent on the reconstruction, rather than just the magnitude of the activation.
>
> > In figure 5 it's not clear to me what the average L0 means -- what is being averaged over here?
>
> The L0 is averaged over a number of input sequences in our dataset. We've now clarified this in the figure caption.
>
> > As far as I understand the SAE objective is non-convex, meaning that there is no guarantee of finding the global optimum. It's therefore also quite possible that different SAEs are simply finding different latents simply because of finding different local minima.
>
> Thank you for raising a valid point about local optima. To investigate this, we have added an extra experiment to Appendix A7 (Figure 19) where we compare the number of reconstruction/novel latents between SAEs of the same size. We find that an average of 94% of latents are reconstruction latents with regard to SAEs of the same size, compared to a maximum of 68% for SAEs of different sizes.

---

> ### Author Response · Authors · 2024-11-20
> **Response 2 to Reviewer pzvM**
>
> > Isn't it clear that (local optima aside) a larger SAE will always find features that are missed by smaller SAE? I'm not sure I follow the argument from line 355 onwards.
>
> While it might seem intuitive that larger SAEs would always find novel features, it wasn't clear whether additional capacity would be used to find new features versus just representing existing features more sparsely. In Figure 2, we provide the “blue square” example of a small SAE that has learned the 6 ground truth features in a dataset, and a large SAE with 9 latents that has learned compositions of those features to further reduce sparsity. Our results empirically demonstrate both effects occur.
>
> > Please add more clarity around treating the latents W_i^{dec} (why is W in boldface)? W_i^{dec} is a scalar quantity, the ith component of the d-dimensional vector W^{dec}. What does it mean to take scalars "training data for our meta-SAE"? The directions W don't convey information about which directions are simultaneously activated. I would have thought it would make more sense to treat the feature vectors f(x) as the entities for learning a meta SAE.
>
> Each decoder direction W_i^{dec} is actually a vector, not a scalar. W^{dec} is a matrix, and we are taking the i'th column, the decoder vector for the i'th SAE latent. The meta-SAE learns to reconstruct these vectors. We have added further clarification of the shapes of these entities in Section 2.
>
> > The BatchTopK function is fully defined. In line 465 it states that the function selects the top K activations, suggesting that this is a mapping from activations to indices. I suspect that the authors mean that the function should return zero for all activations that are not in the top K highest positive values, and is the identity function otherwise.
>
> Yes, your interpretation is correct - BatchTopK zeroes out all activations except the top K highest positive values, for which it acts as the identity function. We've added this explicit definition to Section 6.
>
> > The introduced batch method is used only during training, with a different non-linearity used during "inference". This seems quite strange and it's not clear to me how to justify this. A potential issue not mentioned is that it's quite possible that the batch approach means that some input sequences will have entirely zero activation.
>
> During training, we estimate the threshold on single random batches for efficiency purposes. However, this means that the threshold varies depending on the samples in the batch, resulting in dependencies between samples in the batch. During inference, in order to break this dependency, we use a single threshold estimated over many training batches. This also lets us do inference on small batches (eg a single prompt), where an estimate for the threshold would be very noisy. Our experiments show that this works well. Using different functions during training/inference is common, for example in dropout and batch normalization. If all activations are zero, the reconstruction is equal to the decoder bias, but we have not seen this in practice (see Figure 9). This is the same for ReLU and JumpReLU SAEs, but not for TopK SAEs.
>
> > It's hardly surprising that the BatchTopK SAE has lower SAE than TopK SAE since BatchTopSAE imposes fewer constraints on the objective. I'm not sure why this would be seen as "outperformance".
>
> The goal of papers introducing new SAE training methods (Gao et al. 2024, Rajamanoharan et al. 2024, etc) is to find a method that is better at finding sparse reconstructions, to act as a useful tool for researchers. The standard method is showing better reconstruction performance at similar sparsity levels and number of latents. Our contribution with BatchTopK is to provide a novel method with better performance. We agree that, once thought of, it's unsurprising that BatchTopK is better. Though, ReLU SAEs also impose fewer constraints on the objective than top-k SAEs, but attain worse reconstruction performance Gao et al. 2024
>
> > I'm also unclear as to why BatchTopK SAE is being discussed. Is this method used in all the previous experiments in the paper, or is this a separate piece of work orthogonal to the other contributions of the paper?
>
> BatchTopK was developed specifically to enable training meta-SAEs with very low sparsity (4 active latents per input on average), which existing methods struggled with. It is only used for the meta-SAE experiments and to compare it to existing methods, not in the earlier stitching experiments.
>
> > Figure 11 isn't well explained. Please explain what is being shown here.
>
> Figure 11 (now Figure 12) shows paired examples of latents from GPT2-768 and GPT2-1536 that have high cosine similarity (0.99). For each latent, we show their top activating inputs and the logits they influence most strongly. This demonstrates that similar latents in different sized SAEs capture similar semantic features. We've expanded the text in Appendix A.4 to clarify the figure.

---

> ### Author Response · Authors · 2024-11-20
> **References mentioned in response to Reviewer pzvM**
>
> Samuel Marks, Can Rager, Eric J Michaud, Yonatan Belinkov, David Bau, and Aaron Mueller.
> Sparse feature circuits: Discovering and editing interpretable causal graphs in language models.
> arXiv preprint arXiv:2403.19647, 2024.
>
> Lieberum, T., Rajamanoharan, S., Conmy, A., Smith, L., Sonnerat, N., Varma, V., ... & Nanda, N. (2024). Gemma scope: Open sparse autoencoders everywhere all at once on gemma 2. arXiv preprint arXiv:2408.05147.
>
> Leo Gao, Tom Dupr´e la Tour, Henk Tillman, Gabriel Goh, Rajan Troll, Alec Radford, Ilya
> Sutskever, Jan Leike, and Jeffrey Wu. Scaling and evaluating sparse autoencoders. arXiv preprint arXiv:2406.04093, 2024
>
> Senthooran Rajamanoharan, Arthur Conmy, Lewis Smith, Tom Lieberum, Vikrant Varma, János Kramár, Rohin Shah, Neel Nanda. Improving Dictionary Learning with Gated Sparse Autoencoders. arXiv preprint arXiv:2404.16014, 2024
>
> Trenton Bricken, Adly Templeton, Joshua Batson, Brian Chen, Adam Jermyn, Tom Conerly, Nick
> Turner, Cem Anil, Carson Denison, Amanda Askell, et al. Towards monosemanticity: Decom-
> posing language models with dictionary learning. Transformer Circuits Thread, 2, 2023

---

> ### Author Response · Authors · 2024-11-25
>
> Before this phase of the discussion period ends, we wanted to check in with the reviewer on whether we have addressed your concerns with our work?

---

> > ### Comment · Reviewer_pzvM · 2024-11-25
> >
> > Thanks. Definitely helpful, but I still have some basic concerns.
> >
> > I still don't understand figure 5 and I feel this needs better explanation. The x-axis suggests that the number of features is always increasing, yet the text suggests that features are also swapped (which means a non increase)?
> >
> > For an optimally trained SAE, by definition, using any other features must increase the reconstruction error. I therefore don't understand why say swapping a feature from a small SAE with one from a larger SAE would bring any useful information -- by definition it has to increase the reconstruction error (unless the training got trapped in a suboptimal local optimum). I feel like I'm  missing something important.
> >
> > I also still feel the paper is really for existing believers in this overall approach and doesn't do enough to convince others. For example there is still no explanation early on as to how to associate features with linguistic concepts (which is potentially problematic in itself).
> >
> > Overall, I like the message of the paper, namely that SAEs don't find atomic linguistic units. However, this isn't surprising to me (why should they? -- an LLM is not trained to respect the SAE structure and represent atomic linguistic units).

---

> > > ### Author Response · Authors · 2024-11-25
> > >
> > > Thank you for your response!
> > >
> > > > I still don't understand figure 5 and I feel this needs better explanation. The x-axis suggests that the number of features is always increasing, yet the text suggests that features are also swapped (which means a non increase)? For an optimally trained SAE, by definition, using any other features must increase the reconstruction error. I therefore don't understand why say swapping a feature from a small SAE with one from a larger SAE would bring any useful information -- by definition it has to increase the reconstruction error (unless the training got trapped in a suboptimal local optimum). I feel like I'm missing something important.
> > >
> > > As you say, it is a priori unclear why introducing a latent from a larger SAE into a smaller SAE would improve the reconstruction of that SAE. In particular, prior to our research, it was not known that larger SAEs have entirely novel latents, with respect to a smaller SAE, that do not interfere with the existing latents.
> > >
> > > The reason that reconstruction can improve when modifying a close-to-optimal solution is as follows. While the smaller SAE achieves good performance within its fixed dictionary size, the performance is constrained by its size. Larger SAEs tend to have better reconstruction, all other things being the same. Adding latents from the larger SAE effectively relaxes this dictionary size constraint:
> > >
> > > * Novel latents add previously uncaptured information that the smaller SAE simply didn't have capacity to represent
> > > * During swapping, each swap typically replaces N SAE latents from the smaller SAE with M ≥ N latents from the larger SAE. This larger SAE achieves better reconstruction by having more features available to represent similar information
> > >
> > > So while modifying a solution may worsen performance if we maintained the same dictionary size, here we're actually relaxing the dictionary size constraint. This explains why reconstruction continues improving as we incorporate more latents from the larger SAE's more expressive dictionary.
> > >
> > > The above argument does not imply that any intervention that increases dictionary size can improve reconstruction, just that it is possible. We find that inserting novel latents improves reconstruction while inserting reconstruction latents worsens it. This difference in effects gives us evidence for there being a difference between the two categories of latent.
> > >
> > > > For example there is still no explanation early on as to how to associate features with linguistic concepts (which is potentially problematic in itself).
> > >
> > > Thanks for noting this, we have added the following sentence to the related work section:
> > >
> > > “After training, researchers often interpret the meaning of SAE latents by examining the dataset examples on which they are active, either through manual inspection using features dashboards (Bricken et al., 2023) or automated interpretability techniques (Gao et al., 2024).”
> > >
> > > > Overall, I like the message of the paper, namely that SAEs don't find atomic linguistic units. However, this isn't surprising to me (why should they? -- an LLM is not trained to respect the SAE structure and represent atomic linguistic units).
> > >
> > > We appreciate your enthusiasm for the message of the paper. While there's no a priori reason to expect SAEs to find atomic linguistic units, we believe this assumption has become prevalent in the interpretability community and is popular enough for rigorous evaluation to be of value. SAEs have generated tremendous recent interest, with numerous high-profile papers from top industry and academic labs [1,2,3,4,5,6], media coverage [7, 8], and new startups [9,10]. The influential "Towards Monosemanticity" [1] put forward a vision of SAEs uncovering canonical linguistic features that has since become widespread. Although not all researchers may have believed this initial framing, we still think that explicitly refuting such ideas is valuable and important work. We also believe that the concrete empirical findings we provide through refuting it will advance the community's collective understanding of this popular tool.

---

> > > > ### Author Response · Authors · 2024-11-25
> > > >
> > > > [1] Trenton Bricken, Adly Templeton, Joshua Batson, Brian Chen, Adam Jermyn, Tom Conerly, Nick Turner, Cem Anil, Carson Denison, Amanda Askell, et al. Towards monosemanticity: Decom- posing language models with dictionary learning. Transformer Circuits Thread, 2, 2023
> > > >
> > > > [2] Samuel Marks, Can Rager, Eric J Michaud, Yonatan Belinkov, David Bau, and Aaron Mueller. Sparse feature circuits: Discovering and editing interpretable causal graphs in language models. arXiv preprint arXiv:2403.19647, 2024.
> > > >
> > > > [3] Lieberum, T., Rajamanoharan, S., Conmy, A., Smith, L., Sonnerat, N., Varma, V., ... & Nanda, N. (2024). Gemma scope: Open sparse autoencoders everywhere all at once on gemma 2. arXiv preprint arXiv:2408.05147.
> > > >
> > > > [4] Leo Gao, Tom Dupr´e la Tour, Henk Tillman, Gabriel Goh, Rajan Troll, Alec Radford, Ilya Sutskever, Jan Leike, and Jeffrey Wu. Scaling and evaluating sparse autoencoders. arXiv preprint arXiv:2406.04093, 2024
> > > >
> > > > [5] Senthooran Rajamanoharan, Arthur Conmy, Lewis Smith, Tom Lieberum, Vikrant Varma, János Kramár, Rohin Shah, Neel Nanda. Improving Dictionary Learning with Gated Sparse Autoencoders. arXiv preprint arXiv:2404.16014, 2024
> > > >
> > > > [6] Engels, J., Michaud, E. J., Liao, I., Gurnee, W., & Tegmark, M. (2024). Not all language model features are linear. arXiv preprint arXiv:2405.14860.
> > > >
> > > > [7] https://www.nytimes.com/2024/05/21/technology/ai-language-models-anthropic.html
> > > >
> > > > [8]https://www.technologyreview.com/2024/11/14/1106871/google-deepmind-has-a-new-way-to-look-inside-an-ais-mind
> > > >
> > > > [9] https://goodfire.ai/
> > > >
> > > > [10] https://www.tilderesearch.com

---

> > > > ### Comment · Reviewer_pzvM · 2024-11-25
> > > >
> > > > Thanks. It's clear that by increasing the dictionary size, a reduction in MSE can occur (indeed if b1 were used instead of b0, then as the full SAE1 features are in introduced, then the MSE must reduce to that of SAE1, which must by definition be lower than that of SAE0).
> > > >
> > > > However, figure 5 says "every insertion or *switch* results in a strict improvement in reconstruction (MSE)". This doesn't make sense to me. A switch (which by definition therefore maintains the dictionary size) cannot cause a decrease in an optimal MSE.
> > > >
> > > > Also, the argument that larger dictionary sizes tend to have lower MSEs is specious. This is only (ultimately) guaranteed when the bias b1 of the larger SAE1 is used as discussed above. However, since the bias of the smaller SAE0 is used in the "stitching" method, there is no guarantee of improvement or even expectation of improvement. I feel "stitching" is comparing two unrelated entities and has no obvious significance. This is why I was so curious why you kept the bias b0. The asymmetry of the approach renders comparison of features problematic and is at the heart of my misgivings.

---

> > > > > ### Author Response · Authors · 2024-11-26
> > > > >
> > > > > Thanks! We’re glad you find the effect of dictionary size on MSE in our stitching experiments is clear now.
> > > > >
> > > > > > However, figure 5 says "every insertion or switch results in a strict improvement in reconstruction (MSE)". This doesn't make sense to me. A switch (which by definition therefore maintains the dictionary size) cannot cause a decrease in an optimal MSE.
> > > > >
> > > > > Thank you for raising the point about “every switch” resulting in a “strict improvement”. This was a mistake, and we should have said that on average swaps result in an improvement of reconstruction as is shown in Figure 5. We have updated the caption and text to be more precise. However, in stitching we switch groups of latents between the SAEs where the group of latents from the larger SAE is generally larger than the group of latents from the smaller SAE, see Section 4.2 and examples provided in Appendix A.5. We have modified the caption of Figure 5 to clarify the many-to-many nature of the switches.
> > > > >
> > > > > > Also, the argument that larger dictionary sizes tend to have lower MSEs is specious. This is only (ultimately) guaranteed when the bias b1 of the larger SAE1 is used as discussed above. However, since the bias of the smaller SAE0 is used in the "stitching" method, there is no guarantee of improvement or even expectation of improvement. I feel "stitching" is comparing two unrelated entities and has no obvious significance. This is why I was so curious why you kept the bias b0. The asymmetry of the approach renders comparison of features problematic and is at the heart of my misgivings.
> > > > >
> > > > > The bias terms of the SAEs of different sizes are very similar, and we have previously updated the text of Section 4.1 to clarify this. However, we understand that the cosine similarity of 0.997 indicates some degree of misalignment, and it’s unclear how this might affect stitching experiments. We demonstrate this in Section A.7 Figure 20 by evaluating SAEs of different sizes with their bias terms switched, and observing that this has negligible impact on the performance of the SAEs. As such, the specific SAE from which the bias term in our stitching experiments is sourced will not affect the results of our stitching experiments.
> > > > >
> > > > > As to the concern that stitching is comparing unrelated entities, our two SAEs are two neural networks of the same architecture, trained by the same training procedure, on the same dataset, and differentiated only by their number of latents. We would argue that these are related entities. Comparing even more dissimilar representations with model stitching is common in the literature, e.g. Bansal et al., in which model stitching between neural network model layers is used to compare the representations of different models.
> > > > >
> > > > > Bansal, Yamini, Preetum Nakkiran, and Boaz Barak. "Revisiting model stitching to compare neural representations." Advances in neural information processing systems 34 (2021): 225-236.

---

> > > > > > ### Comment · Reviewer_pzvM · 2024-11-27
> > > > > >
> > > > > > Thanks. I still don't see any substantial improvement of the presentation of figure 5. There are 4 phases in the figure and I cannot find a full description of all of these phases nor why they are chosen and in what order. Why does L0 go up and then down in each of the 4 phases?
> > > > > >
> > > > > > I also disagree with the sentence "This allows us to smoothly interpolate between SAEs of different sizes in terms of dictionary size, sparsity, and reconstruction performance". It's not correct that you're "interpolating" between SAEs. Again, the asymmetry in bias choice means that you cannot interpolate between SAE0 and SAE1.
> > > > > >
> > > > > > Sorry for being a grump here, but the submission still has issues such as: Why bother with a bias term at all? Why is there no recognition that there is a separate challenge of associating which words or concepts will `activate' an SAE feature (I know there is now a reference to how people do this, but it's not recognised as a challenge in itself)? There is no mathematical reason (to my mind at least) to expect the MSE to increase or decrease as latents from the larger SAE are added.  The authors support their results simply because b0 and b1 turn out to be similar. The argument to me it feels like: here is an idea about adding latents from a larger SAE (for which a priori there is no clear reason to expect the MSE to go up or down); however, it turns out that b0 and b1 are similar (empirically) and we'll therefore claim that these results a posteriori have meaning.  Just because others in that community are comparing things that seem even less related doesn't increase my confidence in the rigour of that community.
> > > > > >
> > > > > > I feel the paper addresses a community that somehow already believes in this approach but leaves an outsider like myself scratching my head.  In my view the methodology of using SAEs to find units of meaning in transformers is at best insufficiently well motivated and at worst somewhat meaningless. The paper to some extent supports that view, but I do have a concern about how meaningful the methodology of "stitching" is.
> > > > > >
> > > > > > From the final concluding sentence, what does "leveraging SAEs as flexible tools tailored to the requirements of each analysis" mean?
> > > > > >
> > > > > > I would be happy to raise my score provided that a fuller description of figure 5 is made and there is some attempt to recognise the above concerns.

---

> > > > > > > ### Author Response · Authors · 2024-11-28
> > > > > > >
> > > > > > > Thank you for your continued engagement in helping us improve this paper!
> > > > > > >
> > > > > > >
> > > > > > > > I also disagree with the sentence "This allows us to smoothly interpolate between SAEs of different sizes in terms of dictionary size, sparsity, and reconstruction performance". It's not correct that you're "interpolating" between SAEs. Again, the asymmetry in bias choice means that you cannot interpolate between SAE0 and SAE1.
> > > > > > >
> > > > > > > Thanks for pointing this out. We have updated our methodology and Figure 5 such that when we do the interpolation between two SAEs, we now also interpolate between the biases of the two SAEs by calculating a weighted mean of the bias based on the number of latents from each SAE. In calculating the interpolated bias term, we weight each bias proportionally to the number of latents from each SAE that are included in the stitched SAE. Given that we start with SAE0, end with SAE1, and the points in between are a mix between the two (in terms of latents, biases, reconstruction performance, L0), we believe the term interpolation is now warranted. Given that the biases are so similar, this has no discernable difference on our results, but it helps make our methodology robust to settings where the biases may differ more.
> > > > > > >
> > > > > > > > I still don't see any substantial improvement of the presentation of figure 5. There are 4 phases in the figure and I cannot find a full description of all of these phases nor why they are chosen and in what order. Why does L0 go up and then down in each of the 4 phases?
> > > > > > >
> > > > > > > We have clarified this information in the text introducing Figure 5. There are 4 phases, as we interpolate between five SAEs (size 768 -> 1536, size 1536 -> 3072, etc.). Every colored ‘phase’ corresponds to interpolating between two SAEs (eg. 768 -> 1536), and consists of two parts: first we insert the novel latents (which increases L0), then we swap groups of reconstruction latents (which decreases L0 on average). Novel latents cause the mean L0 to increase, as they only add more latents to the stitched SAE, however reconstruction latents cause the L0 to decrease on average again, as we swap latents in the smaller SAE (eg. blue, circle) with more sparse latents in the larger SAE (eg. blue circle).
> > > > > > >
> > > > > > > > There is no mathematical reason (to my mind at least) to expect the MSE to increase or decrease as latents from the larger SAE are added.
> > > > > > >
> > > > > > > Our study, as with much interpretability research, is empirical rather than mathematical. When we add a latent from a larger SAE to a smaller SAE, the change in MSE tells us something important. If MSE decreases, it means this latent captures information that was completely missing from the smaller SAE. Conversely, if MSE increases, it suggests the smaller SAE was already representing this information in some form, and we're now reconstructing it redundantly. We use this insight to show that smaller SAEs are not complete and do not capture all relevant features from the model activations.
> > > > > > >
> > > > > > > > Why is there no recognition that there is a separate challenge of associating which words or concepts will `activate' an SAE feature (I know there is now a reference to how people do this, but it's not recognised as a challenge in itself)? I feel the paper addresses a community that somehow already believes in this approach but leaves an outsider like myself scratching my head. In my view the methodology of using SAEs to find units of meaning in transformers is at best insufficiently well motivated and at worst somewhat meaningless.
> > > > > > >
> > > > > > > We agree that associating SAE features with concepts is an area of active research and is not yet a solved problem, and have added this caveat to our conclusion. We chose to use standard interpretation techniques rather than tackle this challenge directly. The main message of our work is that even if you accept the standard SAE paradigm for assigning meaning to units, these methods don't learn canonical units. We believe focusing on too many critiques at once would dilute our core message. For researchers who were already skeptical of attributing meaning to SAE features, we believe our work still provides value by deepening their empirical understanding of how SAEs organize information.
> > > > > > >
> > > > > > >
> > > > > > > > From the final concluding sentence, what does "leveraging SAEs as flexible tools tailored to the requirements of each analysis" mean?
> > > > > > >
> > > > > > > What we meant to say here is that “Given that SAEs do not find canonical units of analysis, if one still wants to use SAEs for certain analyses (probing, unlearning, steering) one should adapt the hyperparameters such as dictionary size to find the level of granularity and composition of units that is needed for that specific analysis. We have updated the sentence to make this clearer.

---

> > > > > > > > ### Comment · Reviewer_pzvM · 2024-11-28
> > > > > > > >
> > > > > > > > Thanks. Whilst I still have reservations about the approach, I think that the community of mechanistic interpretabilty will find this an interesting paper. I'll raise my score.

---

### Official Review · Reviewer_jKs2 · 2024-11-03

**Soundness:** 3
**Presentation:** 3
**Contribution:** 4
**Rating:** 8
**Confidence:** 4

**Summary:**

This paper wrestles with the question of whether sparse autoencoders (SAEs) of different sizes learn the same set of "canonical" or "atomic" units. The paper approaches this question from a couple of directions:

1. First, the authors investigate how SAEs of different sizes can be "stitched" together. They find that some features from a larger SAE, when added to a smaller SAE, improve loss (novel latents), and others make loss worse (reconstruction latents). It turns out that there is a rough relationship between, for a given large-SAE latent, the maximum cosine similarity between that latent and the small-SAE latents, and whether adding that latent improve or huts performance. Latents which are dissimilar from all small-SAE latents improve performance of the small-SAE when added, and latents which are quite similar to a small-SAE latent hurt performance of the small-SAE when added. This makes sense -- "novel latents" are features which the small SAE has not learned, and "reconstruction latents" are features which the small SAE has already represented in some manner. The authors describe a procedure by which latents from a large SAE can be added to a smaller SAE -- novel latents can be added, but reconstruction latents must replace latents in the small SAE. This procedure allows one to interpolate between SAEs of different sizes while continuously improving reconstruction error. Nice! Lastly, the authors note that the "reconstruction latents" are not identical to the latents they replace in the smaller SAE, and also sometimes have high cosine similarity with multiple latents in the smaller SAE. One explanation of this is that some large-SAE latents are in fact linear combinations of multiple latents in the smaller SAE. This calls into question whether latents learned by SAEs are atomic.

2. Second, the authors attempt to decompose SAE latents into more atomic features with "meta-SAEs". Meta-SAEs attempt to represent SAE  latents (decoder vectors) as a sparse linear combination of some smaller set of features. Interestingly, the authors report that many meta-SAE features seem interpretable, and many interpretable latents in the original SAE decompose into interpretable combinations of meta-SAE features! For instance, a dedicated "Einstein" latent from the original SAE can be approximated as a linear combination of meta-SAE latents for "Germany", "prominent figures", "scientist", "space and galaxies", and "starts with E". The authors demonstrate that meta-SAE latents are similar to the latents learned by a similarly-sized base SAE.

For their meta-SAEs, the authors trained a new SAE variant called BatchTopK SAEs. While BatchTopK SAEs are not the main contribution of the paper, the authors test them against standard TopK and JumpReLU SAEs, and find that BatchTopK beat TopK SAEs across settings they evaluated, but don't always beat JumpReLU SAEs. BatchTopK SAEs are not the primary contribution of the paper, but are a nice bonus.

**Strengths:**

Overall, the paper addresses an important question about SAE features with reasonable experiments and presentation. This issue, of whether SAE features are "canonical" or "atomic", could have implications for how we scale up SAEs and also for how we use their latents for model interventions, circuit analysis, etc. Overall, the presentation in the paper is clear and the experiments are original and well-executed. Some particular strong points:
* I think Figure 5 is quite compelling. It makes a lot of sense that the L0 would rise as novel latents are added, and it's very cool to see continuous curves like this as one "interpolates" between two SAEs.
* I think that Figure 6 is also quite compelling in showing that meta-SAE latents are pretty similar to similarly-sized base SAE latents.

**Weaknesses:**

* I could not find reported values of the reconstruction loss that meta-SAEs obtain in reconstructing the base 49k-latent SAE latents. How precisely do meta-SAEs actually reconstruct the latents? If they are only a very weak approximation, what would that say about the hypothesis that large-SAE latents are linear combinations of more atomic latents?
* Some minor grammatical and presentation issues: "vertexes" -> vertices, the left quotation marks in the meta-SAE section should be fixed, etc.

**Questions:**

* It's interesting that the reconstruction error falls as much as it does when the reconstruction latents are stitched in. If reconstruction latents were exactly linear combinations of more atomic latents learned by the smaller SAE, I'd expect that replacing those atomic latents with the reconstruction latents would yield the same reconstruction error, but at a lower L0. Instead reconstruction tends to fall *more* steeply during reconstruction latent stitching vs. novel latent stitching. Do you have a guess as to how the reconstruction latents relate to the latents they replace? Perhaps they are a combination of the small-SAE latents but with other additional features included too? I wonder if a feature-manifold explanation might also be worth considering here (Engels et al. 2024), where reconstruction features are more densely covering a feature manifold that corresponding latents in the small-SAE are more coarsely covering. In this sort of model, latents aren't just combinations of atomic latents, and maybe there is no good definition of "atomic latent" when features are multi-dimensional. What do you think?

---

> ### Author Response · Authors · 2024-11-20
> **Response to Reviewer jKs2**
>
> Thank you for your positive and insightful feedback, in particular for recognizing that our work “addresses an important question about SAE features” and that “the presentation in the paper is clear and the experiments are original and well-executed”. We address your main points and questions below:
>
> > I could not find reported values of the reconstruction loss that meta-SAEs obtain in reconstructing the base 49k-latent SAE latents. If they are only a very weak approximation, what would that say about the hypothesis that large-SAE latents are linear combinations of more atomic latents?
>
> Thanks for bringing up this point. The meta-SAE with 2304 meta-latents (and an average L0 of 4) explains 55.47% of the variance of the 49k latents in the SAE. We have included this information in Section 5 of the updated manuscript. Although this only indicates partial linear decomposability, it still provides concrete evidence against the hypothesis that SAEs converge to atomic units. With meta-SAEs, as with SAEs in general, we think the most important result is that any kind of decomposition is possible, not that it is perfect. The fraction of activation variance unexplained for our SAEs ranges from 50% to 10%, in comparison with 45% to 10% for our meta-SAEs, depending on choice of L0 and number of (meta-)latents.
>
> > Some minor grammatical and presentation issues: "vertexes" -> vertices, the left quotation marks in the meta-SAE section should be fixed, etc.
>
> Thank you for pointing these out, they have been fixed in the updated version.
>
> > It's interesting that the reconstruction error falls as much as it does when the reconstruction latents are stitched in.
>
> This is a good observation. The reason that the reconstruction tends to fall more steeply during reconstruction latent stitching vs. novel latent stitching has to do with the order in which we do these. If we swap reconstruction latents before adding in the novel latents, the reconstruction MSE tends to go up a little bit when swapping the reconstruction latents (see Appendix A.7 Figure 17 for the individual effect of swapping reconstruction latents). The steeper fall in MSE during reconstruction latent stitching in Figure 5 occurs because we've already added in the novel latents, which are optimized to work best in combination with the other latents of the larger SAE.
>
> > I wonder if a feature-manifold explanation might also be worth considering here (Engels et al. 2024), where reconstruction features are more densely covering a feature manifold that corresponding latents in the small-SAE are more coarsely covering.
>
> Your hypothesis about the feature manifold explanation from Engels et al. 2024 is interesting. It aligns with our finding that sometimes multiple reconstruction latents have high cosine similarity with multiple small-SAE latents. This could mean that the small SAE makes a lower-dimensional approximation of a higher-dimensional feature manifold that the large SAE represents.  We would be excited to see future work in this direction.

---

> > ### Comment · Reviewer_jKs2 · 2024-11-27
> > **Thanks for answering my questions!**
> >
> > > The meta-SAE with 2304 meta-latents (and an average L0 of 4) explains 55.47% of the variance of the 49k latents in the SAE. We have included this information in Section 5 of the updated manuscript.
> >
> > Thanks for providing this!
> >
> > > The steeper fall in MSE during reconstruction latent stitching in Figure 5 occurs because we've already added in the novel latents, which are optimized to work best in combination with the other latents of the larger SAE.
> >
> > Ah okay, makes sense. For the final version, you'll probably want to increase the resolution of Figure 17.
> >
> > Thanks for answering my questions! I'll keep my score at an 8/10 with a confidence of 4/5. I think this paper makes a solid approach at a basic question in contemporary interpretability work and therefore would be great to include at the conference.

---

### Author Response · Authors · 2024-11-20
**Overall Response**

We thank all reviewers for their thorough and constructive feedback.

We are encouraged that the reviewers found our work addresses "an important question" (jKs2 and 9gei) about “an interesting topic” (pzvM) and that our experiments were assessed as “original and well-executed” (jKs2), “thorough and solid” (9gei), with “detailed experimental information” (HcSs).

Most reviewers found our presentation “clear” (jKs2) and that our “motivation and methods are explained clearly and intuitively, with helpful examples” (HcSs), but we also appreciate the feedback that it was “quite presumptive about readers knowledge of the topic” (pzvM) and 9gei’s dissatisfaction with “how the research is positioned”. We have made the paper more appealing and readable to a broader audience by expanding the explanation of SAEs in Section 2, adding a glossary of terms to the appendix to aid readers with less background knowledge of the field, improving the framing in our introduction, and clarifying the motivation of the BatchTopK section.

Reviewer HcSs and 9gei were both interested in how our work can be used to choose the proper dictionary size for interpretability tasks, such as probing. Prior to our research, it was thought that just training larger SAEs would result in better units of analysis for interpretability tasks. However, our work demonstrates why this is not the case. To reinforce this argument, we have added two sets of interpretability experiments to Appendix A.9, that show the complex relationship between the usefulness of latents for down-stream interpretability tasks and size of the dictionary.

We hope these improvements make our contributions more accessible while preserving the technical rigor appreciated by the reviewers.

---

### Meta-Review · Area_Chair_CsmQ · 2024-12-22

**Metareview:**

This paper examines whether sparse autoencoders (SAEs) discover canonical units of analysis in language models through two novel techniques: SAE stitching, which analyzes relationships between SAEs of different sizes, and meta-SAEs, which attempt to decompose SAE latents into more atomic features. The work also introduces BatchTopK SAEs, a variant that enforces fixed average sparsity.
Reviewers appreciated the paper's thorough empirical analysis of how SAEs organize information, with clear methodology and detailed experimental results. They found the work addresses an important question about the capabilities and limitations of SAEs as interpretability tools.
The main concerns centered on: 1) The positioning of the work specifically for language model interpretability rather than general neural networks, and 2) The initial lack of downstream interpretability experiments to validate the practical implications of the findings. The authors addressed these by clarifying the LLM focus and adding new experiments.

All reviewers recommended acceptance, and I agree.

**Additional Comments On Reviewer Discussion:**

See above

---

### Decision · Program_Chairs · 2025-01-22

Accept (Poster)